

# Opposite variations of peak and low ozone concentrations in eastern China: Positive effects of NOx control on ozone pollution

Zhuang Wang[1,2], Chune Shi[1,2], Hao Zhang[1,2], Xianguang Ji[8], Yizhi Zhu[7], Congzi Xia[10], Xiaoyun Sun[1,2], Xinfeng Lin[2], Shaowei Yan[2], Suyao Wang[9], Yuan Zhou[11,12], Chengzhi Xing[3*], Yujia Chen[1,2*], Cheng Liu[4,3,5,6*]

[1]Anhui Province Key Laboratory of Atmospheric Science and Satellite Remote Sensing, Anhui Institute of Meteorological Sciences, Hefei 230031, China

[2]Shouxian National Climatology Observatory, Huaihe River Basin Typical Farm Eco–meteorological Experiment Field of CMA, Shouxian 232200, China

[3]Key Lab of Environmental Optics and Technology, Anhui Institute of Optics and Fine Mechanics, Hefei Institutes of Physical Science, Chinese Academy of Sciences, Hefei 230031, China

[4]Department of Precision Machinery and Precision Instrumentation, University of Science and Technology of China, Hefei, 230026, China.

[5]Center for Excellence in Regional Atmospheric Environment, Institute of Urban Environment, Chinese Academy of Sciences, Xiamen 361021, China

[6]Key Laboratory of Precision Scientific Instrumentation of Anhui Higher Education Institutes, University of Science and Technology of China, Hefei, 230026, China.

[7]School of Environmental Science and Engineering, Suzhou University of Science and Technology, Suzhou 215009, China

[8]Information Materials and Intelligent Sensing Laboratory of Anhui Province, Anhui University, Hefei 230601, China

[9]Huaibei Meteorological Bureau, Huaibei 235000, Anhui, China

[10]Institute of Big Data for Vocational Education, Guangdong Polytechnic of Science and Technology, Zhuhai 519000, China

[11]Jiangxi Ecological Meteorology Center, Nanchang 330096, China

[12]Nanchang National Climate Observatory, Nanchang 330043, China

*Correspondence to*: Chengzhi Xing (xingcz@aiofm.ac.cn), Yujia Chen (chenyj18@mail.ustc.edu.cn), Cheng Liu (chliu81@ustc.edu.cn)

**Abstract.** Due to the unbalanced emission reductions in ozone precursors in recent years, ozone trends and the causes of their variations in eastern China remain insufficiently understood. To explore the factors controlling ozone trends in eastern China, the long–term ozone precursors observation experiment was conducted. Combined with the satellite and surface measurements, the trend of low, typical and peak ozone concentrations in eastern China was evaluated in detail. Observation results show that the surface peak ozone concentrations significantly decreased (-0.5% per year) while low ozone concentrations increased (0.3% per year) in eastern China during May–September 2017–2022. The underlying cause of surface low and peak ozone trends in eastern China is anthropogenic emission. Ozone formation sensitivity is in VOC–limited regime or transition regime during periods (8:00–11:00) of sharp increases in ozone concentrations, and it is usually in NOx–limited regime when the ozone concentration reaches its peak (~14:00). Substantial reductions in nitrogen oxides (NOx) emissions have diametrically opposed





effects on peak (decreasing) and low (increasing) ozone concentrations, and reducing volatile organic compounds (VOCs) concentrations is the key to reversing the current high ozone level situation in eastern China. In addition, there are obvious interannual variations of surface $O_3$ formation sensitivity on spatial scales through long–term satellite observations, in which area proportion of VOC–limited regime is decreasing while the area proportion of $NO_x$–limited regime is increasing. Our
results highlight the positive impacts of $NO_x$ reduction in controlling peak $O_3$ levels, and regular changes in the ozone formation sensitivity throughout the day should be taken into account when formulating ozone control policies.

## 1 Introduction

In the past decades, China's rapid industrialization and urbanization have brought tremendous economic benefits, but at the same time have exposed us to serious environmental problems (Li et al., 2019; Song et al., 2023; Wang et al., 2023e). Air
pollutants, mainly ozone ($O_3$, warm season) and fine particulate matter ($PM_{2.5}$, cold season), have been the main targets of China's air pollution control (Zhang and Cao, 2015; Xing et al., 2024; Lin et al., 2023; Xing et al., 2022; Bauwens et al., 2022; Wang et al., 2023c). To address the serious air pollution problem, the Chinese government launched the Clean Air Action Plan in 2013 (State Council of China, 2013), which has resulted in the rapid reduction of most air pollutant concentrations. The annual average concentration of $PM_{2.5}$ was reduced by 30% to 50% during 2013–2018 (Zhai et al., 2019), and during 2013–
2017, emissions of nitrogen oxides ($NO_x$) and carbon monoxide (CO) were significantly reduced by 21% and 23%, respectively (Zheng et al., 2018). However, $O_3$ concentrations increased yearly (3.3±4.7 µg/m³/year) over 2015–2019 (Mousavinezhad et al., 2021). Subsequently, the second phase of the Clean Air Action Plan was launched in 2018 (State Council of China, 2018), with new emission controls for $O_3$. Driven by both anthropogenic emissions and meteorological trends, the increasing trend of $O_3$ in eastern China has continued until at least 2019 (Li et al., 2020a), and the formation, sources, and increase of $O_3$ trends
in densely populated areas in eastern China have become an important topic of growing international concern (Wang et al., 2023c; Wang et al., 2023b).

Ozone is produced rapidly in polluted air by the photochemical oxidation of volatile organic compounds (VOCs) in the presence of $NO_x$ ($NO_x = NO + NO_2$) (Li et al., 2022; Cooper et al., 2012). Ambient $O_3$ concentrations are influenced by a variety of factors, including precursor concentrations (Wang et al., 2022; Ding et al., 2023), local meteorological conditions
(Han et al., 2020; Zhai et al., 2019), regional transport (Lang et al., 2021; Wang et al., 2023d), and deposition (Wu et al., 2023). High levels of ambient $O_3$ will have significant impacts on human health, ecosystems, and climate change, and will lead to corresponding severe economic losses (Cheng et al., 2018; Guan et al., 2021; Ge et al., 2021; Gao et al., 2022). The $O_3$ formation sensitivity (VOC–limited regime, transition regime, and $NO_x$–limited regime) prevalent in a region depends on the relative abundance of the two precursors (VOCs and $NO_x$) and their competition for OH radicals, which suggests the
effectiveness of VOCs and $NO_x$ control on $O_3$ production and removal (Wang et al., 2023d; Jin et al., 2020; Ren et al., 2022b). The $NO_x$ concentrations have declined significantly in eastern China since 2013 (Lin et al., 2019), while the increasing trend in anthropogenic VOCs emissions continues until 2019 (Zheng et al., 2018; Bauwens et al., 2022).



Numerous studies have investigated the drivers of the ozone increasing trend in China over the past ten years, especially during the period 2013–2017 (Li et al., 2020a; Liu et al., 2023b; Cheng et al., 2018; Lu et al., 2020; Mousavinezhad et al., 2021). Among those factors, the change of meteorological conditions and anthropogenic emission have been identified as the main causes, with both observations and models coming to similar conclusions that meteorological impacts are not negligible, while anthropogenic impacts dominate the increasing trend of summer $O_3$ in China. For example, Li et al. (Li et al., 2020a) applied a stepwise multiple linear regression model and found that the increase in surface $O_3$ in the North China Plain (NCP) from 2013–2019 was more influenced by changes in emissions (1.2 ppb/year) than meteorological changes (0.7 ppb/year). Furthermore, it has also been found that the decrease in $PM_{2.5}$ is crucial for the increase in $O_3$, which is mainly achieved by reducing the scavenging of hydroxyl ($HO_2$) radicals on the aerosol surface (Li et al., 2019). However, the implications of the heterogeneous uptake of $HO_2$ radicals have been debated. These studies have largely been based on a one–sided understanding of a single ozone indicator, and due to the uneven emission reductions in $O_3$ precursors in recent years, the reasons for the variations in $O_3$ trends in eastern China are not insufficiently investigated, i.e. the background and peak ozone trends. This poses a great challenge for controlling $O_3$ pollution and the development of effective strategies for sustained improvement of air quality.

This study attempts to investigate two issues by combining a large amount of ground–based and satellite observations: (1) to explain the surface ozone background, typical, and peak trends under the uneven emission reductions in $O_3$ precursors in recent years, (2) to explore the key factors triggering ozone pollution in urban areas of eastern China. Firstly, the long–term records of surface $O_3$ and its related parameters observed by urban air quality monitoring sites and satellite in eastern China were reported, and characterize the trends of background, typical and peak surface $O_3$ concentrations during the warm season (May–September) from 2017 to 2022. Then, Multiple Linear Regression Model (MLR) was selected to evaluate the anthropogenic and meteorological contributions to 98th and 2nd $O_3$ percentile trends. Thirdly, formaldehyde (HCHO) and $NO_2$ were employed to diagnosis the diurnal and interannual variations in $O_3$ formation sensitivity and investigate the reasons for the trends in background, typical and peak $O_3$ concentrations in the context of the current unbalanced emission reduction of $O_3$ precursors. Finally, the primary drivers in the $O_3$ exceedances are explored, so as to provide guidance for $O_3$ pollution control strategies and further analyses in eastern China's urban agglomerations.

## 2 Materials and methodology

### 2.1 Surface measurements

The densely populated areas in eastern China mainly contain the NCP and the Middle and Lower Yangtze River Plain (MLYRP), which are vast and economically developed. Considering that ambient $O_3$ production is mainly driven by precursor emissions and meteorological conditions (Wang et al., 2022; Ding et al., 2023; Han et al., 2020; Zhai et al., 2019), the Huaihe River Basin (HRB), which is densely populated with high anthropogenic emissions and climatic vulnerability in China (Jin et



al., 2022; Zhang et al., 2015; Mousavinezhad et al., 2021), was chosen as a major research area. The HRB is located at the junction of Jiangsu, Shandong, Anhui and Henan province, which is one of the regions with the most serious ozone pollution in China (Zhai et al., 2019; Cheng et al., 2018). At the same time, the HRB is also an overlapping area of three transition zones in China (Fig. 1), namely, north–south climate, high–low dimension and Marine–continental transitions, and the northern and

southern parts of the HRB belong to the NCP and the MLYRP, respectively. There are intertwined river network, fertile land, abundant light and heat water resources in the HRB, and it is a common coverage area of the Yangtze River Delta (YRD) economic zone and the Central Plains economic zone, as well as an important grain production base in China (Jin et al., 2022; Zhang et al., 2015; Qiao et al., 2024).

        In this study, $O_3$ trends in 105 cities in eastern China were investigated, mainly focused on 35 cities in the HRB, including

12 cities (Jiaozuo, Xinxiang, Puyang, Kaifeng, Shangqiu, Zhengzhou, Luoyang, Xuchang, Zhoukou, Zhumadian, Xinyang, and Luohe) in Henan province, 6 cities (Zaozhuang, Heze, Jining, Tai'an, Linyi, and Rizhao) in Shandong province, 9 cities (Chuzhou, Suzhou, Bozhou, Fuyang, Benbu, Huainan, Huaibei, Lu'an, and Hefei) in Anhui province, 7 cities (Lianyungang, Suqian, Huai'an, Taizhou, Xuzhou, Yangzhou, and Yancheng) in Jiangsu province, and 1 cities (Suizhou) in Hubei province. The spatial distribution of 35 cities in the HRB is shown in Fig.1a. Note that all the times mentioned in this study refer to local

time.

        Real–time hourly observed urban $O_3$, $NO_2$, and CO concentrations in 105 cities in eastern China from 2017 to 2022 were obtained from the open website of Ministry of Ecology and Environment of China (MEE; https://www.mee.gov.cn; last access: 7 January, 2024), archive at https://quotsoft.net/air/ (last access: 7 January, 2024). As of 31 August, 2018, MEE reported concentrations in µg/m³ under standard conditions (273 K, 1013 hPa). The reference status was changed to 298 K and 1013

hPa on 1 September, 2018. To facilitate analysis of the long–term series, the mass concentrations (µg/m³) of $O_3$, $NO_2$, and CO at each site were converted to volume mixing ratios (VMRs, ppb) to eliminate the effect of these changes on trend calculations. Data quality control methods and calculation of daily maximum 8–hour $O_3$ (MDA8 $O_3$) concentrations are implemented according to the data statistical requirements of the Technical Regulations for Environmental Air Quality Evaluation (Trial) (HJ633–2013). According to the Technical Regulations of China Ambient Air Quality Index (Trial) (HJ633–2012), an MDA8

$O_3$ concentration greater than 160 µg/m³ is defined as $O_3$ exceedance days, otherwise it is $O_3$ normal day.

### 2.2 MAX–DOAS measurements

        Three typical cities in the 35 cities in the HRB were selected to be representative of the whole basin to conduct Multi–Axis Differential Optical Absorption Spectroscopy (MAX–DOAS) observations, namely Hefei, Huaibei and Tai 'an. Firstly, the three cities are located in similar longitudes with large differences in latitudes, transitioning sequentially from south to

north (Fig. 1a). Secondly, the $O_3$ concentrations of the three cities differed greatly, with Tai' an having a higher mean MDA8 $O_3$ concentration (ranked second), Hefei having a lower mean MDA8 $O_3$ concentration (ranked 32th), and Huaibei having an intermediate MDA8 $O_3$ concentrations (ranked 15th) among all the cities in the HRB. The three observation stations in Tai' an



(36.205 °N, 117.094 °E), Huaibei (33.962 °N, 116.805 °E) and Hefei (31.827 °N, 117.233 °E) are all set up in urban areas, and the observation periods are from 15 July 2021 to 15 May 2023, 12 April 2019 to 27 May 2022, and 22 December 2020 to 15 May 2023, respectively.

The MAX–DOAS employed in this study consists of a telescope, two spectrometers (UV: 303–370 nm; VIS: 390–550
nm, temperature stabilized at ~20°C), and a computer as a control and data acquisition unit. The telescope elevation angles were set to 1–6, 8, 10, 15, 30, and 90°, controlled by stepper motors. MAX–DOAS measured spectral information to retrieve aerosols and trace gas profiles. The system was operated only during daytime (08:00–17:00 local time) with a temporal resolution of 15 minutes and a spatial resolution of 100 m, and since this study focuses on the surface $O_3$ formation sensitivity, only the lowest level data of $NO_2$ and HCHO VMRs (ppb) are used. Detailed information on the MAX–DOAS instrument,
measurement procedures, data inversion algorithms, and data quality control can be found in our previous studies (Xing et al., 2023; Gao et al., 2023; Liu et al., 2021b; Liu et al., 2022a; Liu et al., 2021a; Wang et al., 2020).

The comparison of ground–based MAX–DOAS and Tropospheric Monitoring Instrument (TROPOMI) observations was conducted to ensure the reliability of the data used in this study. Since the satellite overpass was around 13:30, the mean MAX–DOAS results between 13:00 and 14:00 was taken for comparison. TROPOMI observations were averaged over a range of
0.2° from the ground–based MAX–DOAS station. Comparisons of $NO_2$ and HCHO tropospheric vertical column density (VCD) from MAX–DOAS and TROPOMI observations are shown in Fig. S1 and S2. In general, the MAX–DOAS and TROPOMI observations are in high agreement. The Pearson correlation coefficient of monthly average $NO_2$ VCD is 0.99 (P<0.01), 0.96 (P<0.01) and 0.96 (P<0.01) in Hefei, Huaibei and Tai' an, respectively, and the Pearson correlation coefficient of monthly average HCHO VCD is 0.88 (P<0.01), 0.77 (P<0.01) and 0.68 (P<0.01) in Hefei, Huaibei and Tai' an, respectively.
Generally, $NO_2$ and HCHO VCD observed by TROPOMI are smaller than those observed by MAX–DOAS. This is because the TROPOMI sensitivity peaks in the upper troposphere, whereas the MAX–DOAS sensitivity usually peaks at or near the surface (De Smedt et al., 2021; Dimitropoulou et al., 2020), and the $NO_2$ and HCHO concentrations usually concentrated in the boundary layer over the polluted city (Fig.S3). In addition, the bottom $NO_2$ concentrations observed by MAX–DOAS were also compared with urban surface $NO_2$ concentrations measured by MEE (Fig.S4), and the results were also comparable to the
comparisons reported in previous studies (Lin et al., 2022; Wang et al., 2020), with Pearson correlation coefficient of 0.74 (P<0.01), 0.66 (P<0.01), and 0.73 (P<0.01) for Hefei, Huaibei, and Tai' an, respectively.

## 2.3 Satellite observations

The TROPOMI is an imaging Spectrometer on board the European Space Agency's Copernicus Sentinel 5 Precursor satellite, launched in October 2017 with a daily overpass of about 13:30. TROPOMI has a spatial resolution of 3.6 × 7.2 km
(before 6 August 2019) and 3.6 × 5.6 km (after 6 August 2019). A more detailed description of the TROPOMI can be found in Veefkind et al. (Veefkind et al., 2012). The TROPOMI $NO_2$ ("S5P_OFFL_L2__NO2…") and HCHO ("S5P_OFFL_L2__HCHO…") tropospheric VCD products during May–September 2018–2022, and TROPOMI $O_3$ profiles



("S5P_OFFL_L2__O3__PR…") during May–September 2022 were used in this study (Download from https://search.earthdata.nasa.gov/search; last access: 7 January, 2024) (Van Geffen et al., 2020; De Smedt et al., 2018). The recommended quality control (QC, in the range of 0–1) filter was applied to exclude HCHO retrieval values with QC marks less than 0.5, $NO_2$ retrieval values with QC marks less than 0.75, and surface $O_3$ retrieval values with QC marks less than 0.5.

In addition, the TROPOMI observed HCHO VCD and $NO_2$ VCD were regridded to $0.05 \times 0.05°$ (about $5 \times 5$ km), and surface $O_3$ VMRs were regridded to $0.25 \times 0.25°$ (about $25 \times 25$ km) in this study.

**2.4 Stepwise Multiple Linear Regression Model**

To quantify the importance of meteorological drivers, numerous previous studies have used stepwise MLR to derive relationships between meteorological factors and observed surface $O_3$ concentrations in China (Li et al., 2019; Liu et al., 2023b;

Li et al., 2020a; Zhang et al., 2023b; Han et al., 2020). These studies have demonstrated the important skill of stepwise MLR in quantifying the contribution of meteorological and anthropogenic components of $O_3$ concentrations. The stepwise MLR model takes the following form:

$$y = R + \sum_{d=1}^{n} \alpha_d Met_d$$

(1)

Where y is observed surface $O_3$ VMRs, R is the regression constant, $\alpha_d$ is the regression coefficient, and $Met_d$ is the

meteorological fields considered as possible $O_3$ covariates.

Firstly, the stepwise MLR model was used to select the key meteorological parameters for the variation of daily 98[th] and 2[nd] $O_3$ percentiles. Normalized meteorological variables were obtained from the ERA5 reanalysis data (Download from https://cds.climate.copernicus.eu, last access: 7 January, 2024), included temperature (T, °C), surface relative humidity (RH, %), total cloud cover (TCC), UV radiation at the surface (UVB, w/m$^2$), total precipitation (TP, mm), mean sea level pressure (MSLP,

hPa), wind speed of u and v components (U, V, m/s), boundary layer height (BLH, m), and vertical velocity at 850 hPa (V850, m/s). The details of the meteorological parameters are shown in Table S1. To avoid overfitting, only the three most important meteorological parameters were selected (Li et al., 2020a; Li et al., 2019), and they were also required to be statistically significant above the 95% confidence level in the MLR model.

The stepwise MLR model calculated daily 98[th] or 2[nd] $O_3$ percentile concentrations represent baseline $O_3$ concentrations

perturbed by meteorological variables, which defined as the contribution of the meteorological component ($O3_{mete.}$). The observed $O_3$ VMRs ($O3_{obs.}$) minus the meteorological component is considered the anthropogenic component ($O3_{anth.}$). Assuming that $O_3$ precursor emissions from natural sources are relatively stable (Zhang et al., 2023b), observed $O_3$ VMRs can be considered as the natural background concentration ($O3_{bg}$) added to the perturbations caused by meteorological ($\Delta O3_{mete.}$) and anthropogenic emissions ($\Delta O3_{anth.}$):

$$O3_{obs.} = O3_{bg} + \Delta O3_{mete.} + \Delta O3_{anth.}$$  (2)



The natural background $O3_{bg}$ can be represented by the long–term (2017–2022) mean values of the stepwise MLR derived $O3_{mete.}$ ($\overline{O3_{mete.}}$), and the meteorological perturbation can be represented by the anomaly in the stepwise MLR derived $O3_{mete.}$:

$$\Delta O3_{mete.} = O3_{mete.} - \overline{O3_{mete.}} \tag{3}$$

Therefore, the observed $O_3$ anomalies ($\Delta O3_{obs.}$) can be calculated as.

$$\Delta O3_{obs.} = O3_{obs.} - \overline{O3_{mete.}} \tag{4}$$

From Eqs.2 and 4, the observed $O_3$ anomalies are determined by perturbations from meteorological and anthropogenic influences:

$$\Delta O3_{obs.} = \Delta O3_{mete.} + \Delta O3_{anth.} \tag{5}$$

**2.5 Regression Model for source separation in primary and secondary HCHO**

Tracer–driven linear regression models are widely used to separate primary and secondary sources of HCHO (Hong et al., 2018; Lin et al., 2022; Liu et al., 2023a; Sun et al., 2021; Bao et al., 2022; Heue et al., 2014; Macdonald et al., 2012). CO is emitted directly into the atmosphere through combustion processes (e.g., incomplete combustion in vehicle engines) and can be used as a tracer for primary HCHO emissions (Garcia et al., 2006). $O_3$ reacts with NO emitted from automobiles to form $NO_2$. Therefore, odd oxygen ($O_x = O_3 + NO_2$) is often used as a tracer of photochemical processes in urban atmospheres (Wood et al., 2010). In this study, CO and $O_x$ were selected as tracers to separate the primary and secondary sources of ambient HCHO as shown in Equation 6:

$$[HCHO] = \beta_0 + \beta_1 \times [CO] + \beta_2 \times [O_x] \tag{6}$$

Where $\beta_0$, $\beta_1$, and $\beta_2$ are the regression coefficients. [HCHO], [CO], [$O_x$] are the ambient HCHO, CO and $O_x$ VMRs, respectively.

The relative contributions of primary, secondary and atmospheric background HCHO to total ambient HCHO were calculated from the tracer VMRs and the corresponding regression coefficients:

$$P_{bg} = \frac{\beta_0}{\beta_0 + \beta_1 \times [CO] + \beta_2 \times [O_x]} \times 100\% \tag{7}$$

$$P_{pri} = \frac{\beta_1 \times [CO]}{\beta_0 + \beta_1 \times [CO] + \beta_2 \times [O_x]} \times 100\% \tag{8}$$

$$P_{sec} = \frac{\beta_2 \times [O_x]}{\beta_0 + \beta_1 \times [CO] + \beta_2 \times [O_x]} \times 100\% \tag{9}$$

Where $P_{pri}$ denotes the contribution of primary sources HCHO (e.g., vehicular and industrial emissions), $P_{sec}$ denotes the contribution of secondary sources HCHO (photochemical oxidation), and $P_{bg}$ denotes background HCHO. According to previous studies in central and eastern China (Ma et al., 2016; Wang et al., 2015), the background level of HCHO was limited



to 1 ppb. Therefore, the regression parameter $\beta_0$ was fixed at 1 ppb in this study (Hong et al., 2018; Lin et al., 2022). The fitted parameters of the MLR for measured and modelled HCHO are shown in Fig.S5.

## 3 Results and discussions

### 3.1 Trends of O₃ and its precursors in HRB

5   Figure 1 presents the spatial distributions of surface mean MDA8 $O_3$ VMRs at all available urban sites in HRB during May–September 2017–2022. High MDA8 $O_3$ VMRs are concentrated in the northwest cities of the HRB, with the highest values in Jiaozuo, up to 86 ppb. High $O_3$ VMRs indicate intensive anthropogenic emissions of $O_3$ precursors in these cities, most of which are located on the NCP, one of the most polluted areas in China (Li et al., 2019). Fig. 1b and c show the average VCDs of $NO_2$ and HCHO observed by TROPOMI during May–September 2018–2022, respectively. High $NO_2$ and HCHO

10 level are mainly clustered in the Beijing–Tianjin–Hebei (BTH) and Yangtze River Delta (YRD) regions, while the northern and southern parts of the HRB belong to the BTH and the YRD, respectively. The $O_3$ and its precursors level of HRB is often affected by these two regions (Song et al., 2023). In addition, the two lake basins (TWB) in the southwestern part of the HRB also have high $O_3$ precursors emissions, which may also have an impact on $O_3$ pollution in the HRB. Therefore, the HRB is also located on the transport path of air pollutants from several major economic zones in China, and its $O_3$ concentration is not

15 only controlled by local emissions and meteorological conditions, but also needs to consider the adverse effects of the transport of $O_3$ and its precursors.

   In this study, the 2[nd], 50[th], and 98[th] percentiles of hourly $O_3$ and $NO_2$ concentrations in each month for cities in the HRB were calculated to determine their long–term trends at low, typical, and peak concentration levels (Li et al., 2022; Cooper et al., 2012; Gaudel et al., 2020). Different percentiles may be related to different influences, such as background concentration

20 levels, emission changes, climate change and regional transport effects (Lefohn et al., 2010). Li et al. (Li et al., 2014) have reported that concentrations below 5[th], between 25[th] and 75[th], and above 95[th] represent background, typical and polluted concentrations, respectively. Using these indicators to investigate long–term changes in $O_3$ and $NO_2$ helps to avoid one–sided understanding resulting from analyses using a single type of statistical data.

   As shown in Fig.2, the 98[th] $O_3$ percentile in the HRB showed a significant decreasing trend (about -0.43 ppb/year, -0.5%

25 per year) during May–September 2017–2022, and the corresponding $O_3$ exceedance days (-1.5% per year) and exceedance hours (-1.9% per year) are also dropped. This is in stark contrast to previous studies, which have widely observed a rapid increase in average surface $O_3$ concentrations in Chinese cities while ignoring changes in their peak concentrations (Li et al., 2021b; Li et al., 2019; Lu et al., 2020; Chan et al., 2017; Liu et al., 2023b). The decline rate of 50[th] $O_3$ percentile is considerably slower than that of 98[th] $O_3$ percentile during May–September 2017–2022, about -0.04 ppb/year (-0.1% per year). Unexpectedly,

30 the 2[nd] $O_3$ percentile in the HRB showed an increasing trend (about 0.05 ppb/year, 0.2% per year) during May–September



2017–2022, the increase in $2^{nd}$ $O_3$ concentrations may be due to the decrease in $O_3$ titration from fresh NO emissions as $NO_x$ emissions decrease (Li et al., 2023). It can be confirmed in Fig. 2b that all the three $NO_2$ percentiles ($2^{nd}$, $50^{th}$, and $98^{th}$) show a significant decreasing trend during May–September 2017–2022, with relative decreasing tends of -2.2%, -2.2%, and -2.5% per year, respectively. Notably, $NO_2$ declines more rapidly in 2020 and 2021, mainly due to the impact of the COVID-19

pandemic, which has been discussed in detail in previous studies (Liu et al., 2022b). Satellite observations also found a significant decrease in $NO_2$ concentration in the HRB (Fig.3a), with a decrease rate of about $0.07 \times 10^{15}$ mol/cm$^2$/year  (-1.2% per year). Since 2013, there has been a significant decrease in $NO_x$ emissions in China due to a series of air pollution prevention and control policies (Wang et al., 2019; Lin et al., 2019).

We further investigated the trends of $98^{th}$, $50^{th}$ and $2^{nd}$ $O_3$ percentiles, $O_3$ exceedance hours, and mean $NO_2$ concentrations

in each city of HRB during May–September 2017–2022 (Table S2). All cities in the HRB showed significant decreasing trends in mean $NO_2$ concentrations, and the peak $O_3$ concentrations and $O_3$ exceedance hours are decreasing in almost all cities, while the $2^{nd}$ $O_3$ percentiles is increasing. Therefore, the trends of $O_3$ and $NO_2$ concentrations for each city in the HRB are almost identical. In addition, we also examined the relationship between the trend of mean $NO_2$ concentrations and the trend of $98^{th}$ $O_3$ percentiles form May–September 2017–2022, which showed a significant positive correlation, with a Pearson correlation

R=0.53 (P<0.01; Fig.4). In contrast, there is no significant correlation between the trend of mean $NO_2$ concentrations and the trend of $2^{nd}$ $O_3$ percentiles during May–September 2017–2022, meteorological factors may have some influence on the trend of $2^{nd}$ $O_3$ percentiles concentrations, which will be discussed in detail in section 3.2. However, in general, with the decreasing of $NO_x$ concentrations, the $O_3$ titration from fresh NO emissions as $NO_x$ emissions will be weakened (Li et al., 2023), resulting in an increase in the $2^{nd}$ $O_3$ percentile concentrations. Therefore, the significant reduction of $NO_x$ emissions may have had

opposite effects on the low and peak $O_3$ levels.

Due to the lack of long–term ground–based VOCs observations, Satellite observed HCHO VCD was used to indicate VOCs variations in the HRB during May–September 2018–2022. HCHO is a transient oxidation product of a variety of VOCs, reflecting long–term trends in active VOCs, and changes in satellite HCHO can roughly indicate VOCs emission variations, which has been applied in a number of previous studies (Zhang et al., 2019; Zheng et al., 2018). The HCHO VCD observed

by TROPOMI in the HRB showed a significant upward trend during May–September 2018–2022, about 0.14 $\times 10^{15}$ mol/cm$^2$/year  (0.9% per year) (Fig.3b). In general, HCHO VCD varies with the seasons (Li et al., 2021a), high temperature and sufficient radiation are conducive to photochemical reactions of VOCs and plant growth, which in turn released more biogenic VOCs, promoting the formation of HCHO (Ren et al., 2022b). In China, anthropogenic VOCs emissions continue to increase until 2019 (Zheng et al., 2018; Bauwens et al., 2022), then, anthropogenic VOCs emissions

were brought under control. However, biogenic VOC emissions have increased in recent years, i.e., record–breaking heat swept across the HRB in 2022, and biogenic VOCs emissions increased dramatically (Zhang et al., 2023a; Zhang et al., 2022). More importantly, reductions in anthropogenic VOCs have not been sufficient to reduce total VOC emissions (Li et al., 2020b). Therefore, the overall trend of VOCs emissions in the HRB is increasing slowly.



### 3.2 Anthropogenic and meteorological contributions to O₃ trends

The MLR model was applied to assess the significance of anthropogenic and meteorological components on 98$^{th}$ and 2$^{nd}$ O₃ percentile trends. It is worth noting that the aim of this section is not to accurately assess the contribution of each meteorological factor and precursor concentrations to O₃ trends, but to explore variations in 98$^{th}$ and 2$^{nd}$ O₃ percentile

concentrations due to variations in meteorological conditions and anthropogenic emissions in the HRB, and to determine the magnitude of 98$^{th}$ and 2$^{nd}$ O₃ percentile trends. Based on the daily output of the MLR model, the monthly mean meteorological and anthropogenic components of 98$^{th}$ and 2$^{nd}$ O₃ percentiles from May–September 2017–2022 were calculated, and the relative contributions of the meteorological and anthropogenic components to the 98$^{th}$ and 2$^{nd}$ O₃ percentiles trends were quantified. According to the MLR model, the three key meteorological factors with the most significant effects on 98$^{th}$ and 2$^{nd}$

O₃ percentiles were selected (Li et al., 2020a; Li et al., 2019). Then, the meteorological components of 98$^{th}$ and 2$^{nd}$ O₃ percentile concentrations were simulated using the MLR model driven by the three selected key meteorological factors, and the coefficients of each factor and the Pearson correlation coefficients (R) of the fitted MLR model were obtained. The R for each city of the HRB was in the range of 0.37 to 0.58 for 2$^{nd}$ O₃ percentiles (Fig.S6) and 0.56 to 0.77 for 98$^{th}$ O₃ percentiles (Fig.S7), respectively. RH is the most important factor affecting 98$^{th}$ and 2$^{nd}$ O₃ percentiles concentrations, followed by T, BLH, TCC,

U, V, and V850 (Table S3 and S4). These results are basically consistent with the current understanding of the meteorological effects of O₃ (Li et al., 2020a; Weng et al., 2022; Ding et al., 2023).

Figure 5 shows the variations in monthly mean observed 98$^{th}$ and 2$^{nd}$ O₃ percentiles concentrations, meteorological 98$^{th}$ and 2$^{nd}$ O₃ percentiles component in MLR simulations, and the anthropogenic 98$^{th}$ and 2$^{nd}$ O₃ percentiles component during May–September 2017–2022. The interannual fluctuation of the mean 98$^{th}$ O₃ percentiles meteorological component obtained

by the MLR model from May–September 2017–2022 is considerably smaller than that of the anthropogenic component, with almost no trend between 2017 and 2022 (0.003 ppb/year). Meteorological influences explain only 1.0% (0.003 ppb/year) of the observed 98$^{th}$ O₃ percentiles trend during May–September 2017–2022, with the remaining -101% (-0.299 ppb/year) determined by anthropogenic influences. The interannual fluctuation of the mean 2$^{nd}$ O₃ percentiles meteorological component (-0.013 ppb/year, -14%) is also smaller than that of the anthropogenic component (0.108 ppb/year, +114%). Therefore, the

effect of the change of meteorological component on 98$^{th}$ O₃ percentile trends is almost negligible, but has some influence on the 2$^{nd}$ O₃ percentile trends. Overall, the 98$^{th}$ and 2$^{nd}$ O₃ percentile trends in HRB mainly depend on anthropogenic emissions.

It is worth noting that the contribution of meteorological components on 2$^{nd}$ O₃ percentiles trends is -0.013 ppb/year, while the overall trend of 2$^{nd}$ O₃ percentiles is 0.095 ppb/year. A recent study shows that nighttime O₃ depletion in China is mainly caused by wet scavenging effect and O₃ titration from fresh NO emissions (Li et al., 2023). The wet scavenging effect

is similar to the effect of precipitation, the higher the ambient humidity, the more conducive to O₃ depletion. The RH at nighttime (19:00–07:00) increased slowly in the HRB during May–September 2017–2022 (Fig.S8a), and a general wetting trend can also be detected in the HRB during summer in recent years (Hu et al., 2021). RH had the most significant effect on 2$^{nd}$ O₃ percentile trends according to MLR results (Table S4). Therefore, the meteorological component has an inhibitory effect





on the increase of $2^{nd}$ $O_3$ percentile trends, but due to the significant emission reduction of $NO_x$ concentrations, the titration of $NO_x$ is weakened (0.108 ppb/year, +114%), the decrease of $O_3$ depletion at night leads to the increase of the overall $2^{nd}$ $O_3$ percentile trends.

**3.3 Diurnal differences in surface $O_3$ formation sensitivity**

In order to elucidate in detail the reasons for changes in peak, typical, and background $O_3$ concentrations in the presence of uneven emission reductions in $O_3$ precursors (substantial decrease of $NO_x$ emissions and slightly increase of VOCs emissions), it is critical to accurately determine the $O_3$ formation sensitivity to its precursors, especially at high temporary resolution. Here, secondary HCHO and $NO_2$ are intended to be selected as representatives of VOCs and $NO_x$ respectively (Hong et al., 2018; Ren et al., 2022a; Xue et al., 2022; Hong et al., 2022; Lin et al., 2022). The surface $NO_2$ and HCHO VMRs

retrieved from ground–based MAX–DOAS observations were used to diagnose the surface $O_3$ formation sensitively regime. The thresholds for VOC–limited regime, Transition regime, and $NO_x$–limited regime were determined based on the correlation between the $O_3$ concentrations and the changes in $O_3$ precursor concentrations under different FNR (defined as the ratio of HCHO VMRs to $NO_2$ VMRs, FNR = HCHO/$NO_2$). Here, we use $FNR_{sec}$ (defined as the ratio of secondary HCHO to $NO_2$, $FNR_{sec}$ = $HCHO_{sec}/NO_2$) as an indicator of $O_3$ formation sensitivity. Compared with conventional FNR, $FNR_{sec}$ eliminates

background and primary HCHO interference, improves the accuracy of diagnosing $O_3$ formation sensitivity, and contributes to a better understanding of $O_3$ formation sensitivity (Xue et al., 2022; Lin et al., 2022). Most of $FNR_{sec}$ values (~98%) varied between 0.03 ~ 1.5 during the whole observations, and excessively low $FNR_{sec}$ values can be attributed to deficiencies in the HCHO source assignment in the MLR model. Therefore, $FNR_{sec}$ values greater than 1.5 or less than 0.03 were filtered out for quality control.

There are three steps to determine the $FNR_{sec}$ threshold. Firstly, the surface secondary HCHO and $NO_2$ VMRs during the whole observation period were normalized by dividing their respective mean values because of the large differences in surface HCHO and $NO_2$ concentrations (Ren et al., 2022a; Su et al., 2017). The linear fitting slope of $O_3$ VMRs with normalized secondary HCHO VMRs and normalized $NO_2$ VMRs were then calculated, respectively. Finally, third order polynomials were used to fit the slopes of $O_3$ VMRs versus normalized $NO_2$ VMRs ($S_{NO2}$) and the slopes of $O_3$ VMRs versus normalized

secondary HCHO VMRs ($S_{HCHO}$), respectively (Fig. 6).

As shown in Fig.6, Three–order fitting of $S_{NO2}$ increases almost linearly with $FNR_{sec}$ values, as do the $S_{HCHO}$. When $S_{NO2}$ is much larger than $S_{HCHO}$, $O_3$ formation is more sensitive to $NO_x$, which is $NO_x$–limited regime, and vice versa. For example, in Hefei, $S_{NO2}$ and $S_{HCHO}$ intersect at $FNR_{sec}$=0.21. $FNR_{sec}$ less than 0.16 and greater than 0.29 correspond to VOC–limited regime and $NO_x$–limited regime, respectively, where the relative difference between $S_{NO2}$ and $S_{HCHO}$ is more than 25% (Lin et

al., 2022), and the range of $FNR_{sec}$ from 0.16 to 0.29 represents a transition regime. For Huaibei and Tai'an, the transition regime range was 0.24–0.44 and 0.14–0.24, respectively. The $FNR_{sec}$ threshold of $O_3$ formation sensitivity varies in different cities, which may be caused by the difference in $O_3$ precursors emissions.



Figure 7 shows the diurnal variation trend of $O_3$ and its precursors at three stations (Hefei, Huaibei, and Tai'an) in the HRB from south to north. $O_3$ and $NO_2$ VMRs showed diametrically opposite trends from 08:00 to 13:00, with $O_3$ concentrations rapidly increasing (about 8.2, 7.4, and 8.6 ppb/hour in Hefei, Huaibei, and Tai' an, respectively) and $NO_2$ concentrations gradually decreasing (about -0.90, -0.86, and -1.49 ppb/hour in Hefei, Huaibei, and Tai' an, respectively). Ambient HCHO

concentrations depend on primary emissions and photo–oxidation of VOCs (Xue et al., 2022). We separated the primary and secondary sources of HCHO using CO and $O_x$ VMRs, and their diurnal variations are shown in Fig. 7 d–f. The primary source contributes the most to the ambient HCHO concentrations. The HCHO from primary emissions was highest between 08:00 and 10:00 (about 2.63, 2.50, and 4.30 ppb in Hefei, Huaibei, and Tai' an, respectively), then gradually decreased, reaching the lowest concentration around 15:00, about 1.78, 1.54, and 1.82 ppb in Hefei, Huaibei, and Tai' an, respectively, and then

gradually rising. The high primary HCHO concentrations in the morning and evening may be due to emissions from traffic (Zhang and Cao, 2015; Lin et al., 2022; Hong et al., 2018). Secondary HCHO concentrations were lowest in the morning, about 0.58, 0.84, and 0.65 ppb in Hefei, Huaibei, and Tai' an, respectively, with the enhancement of photochemical reactions and the resumption of human activities, the secondary HCHO concentrations gradually increased from 08:00 to 12:00, and the first peak usually occurred at 11:00–14:00. The proportion of secondary HCHO VMRs in the total HCHO VMRs also increased

rapidly, and the proportion of secondary HCHO VMRs in the total HCHO VMRs gradually stabilized after 12:00, about 24%, 31%, and 22% in Hefei, Huaibei, and Tai' an, respectively. Similar diurnal variation trend also found in Shenyang (Xue et al., 2022), Nanjing (Hong et al., 2018), Guangzhou (Lin et al., 2022) and Shenzhen (Zhang and Cao, 2015), and also in Rome (Possanzini et al., 2002), Mexico (Possanzini et al., 2002), and Toyama (Taguchi et al., 2021), etc. It is worth noting that secondary HCHO VMRs in Hefei, Huaibei and Tai' an increased significantly between 16:00 and 17:00, and the specific

reasons remain to be further investigated. In general, $NO_2$ concentrations were higher and secondary HCHO concentrations were lower in the early morning, and with the enhancement of photochemical reactions (08:00 to 13:00), $NO_2$ concentrations decreased rapidly and secondary HCHO concentrations increased gradually.

The significant diurnal variation of $O_3$ precursors also led to the transition of $O_3$ formation sensitivity, with $FNR_{sec}$ demonstrating a significant single–peak characteristics. As shown in Fig. 7 j–l, $FNR_{sec}$ increases rapidly from 08:00 to 13:00,

when $O_3$ VMRs were also considerably increased. The $O_3$ formation sensitivity of Hefei, Huaibei and Tai' an were in VOC–limited regime (below the shaded area) at 08:00 and 09:00, and gradually shifted to transition regime at 10:00 and 11:00. From 12:00 to 14:00, the $O_3$ formation sensitivity in Huaibei was basically in $NO_x$–limited regime, and then gradually shifted to transition regime after 15:00. The $O_3$ formation sensitivity of Hefei and Tai' an were in $NO_x$–limited regime from 12:00 to 16:00, and then gradually turned to transition regime after 17:00. We further investigated the diurnal variation characteristics

of $FNR_{sec}$ on $O_3$ exceedance days (MDA8 $O_3$>160 µg/m$^3$). Compared with the whole observation period, the $FNR_{sec}$ on $O_3$ exceedance days shifted faster from 08:00 to 13:00, and stays in the $NO_x$–limited regime for a longer period of time. Ren et al. (Ren et al., 2022a) have also conducted a short–term observation experiment of $O_3$ formation sensitivity based on lidar and MAX–DOAS observations in September 2020 in Hefei, and found the similar $O_3$ formation sensitivity transitions.

The diurnal variation of $O_3$ VMRs was very similar to that of $FNR_{sec}$, especially from 08:00 to 13:00, both of which





increased sharply. The exponential function was applied to fit the relationship between $O_3$ concentration and $FNR_{sec}$ values from 08:00 to 13:00 in Hefei, Huaibei, and Tai' an, respectively (Fig.8), and all three cities showed significant positive correlations, the Pearson correlation coefficients are 0.53 (p<0.01), 0.40 (p<0.01), and 0.52 (p<0.01) in Hefei, Huaibei, and Tai' an, respectively. Moreover, the exponential fitting is better on the $O_3$ exceedance days, and the correlation coefficients are

higher, 0.69 (p<0.01), 0.59 (p<0.01), and 0.61 (p<0.01) in Hefei, Huaibei, and Tai' an, respectively. It indicates that the dependence of $O_3$ production rate on its precursors is rapidly shifting with increasing $O_3$ concentrations, especially during $O_3$ exceedance days. These changes also suggest that the dependence of $O_3$ on its precursor is extremely complex, precise control of $O_3$ pollution requires the identification of $O_3$ formation sensitivity mechanisms with high temporal resolution, and targeted control of $O_3$ precursors concentrations.

**3.4 Interannual differences in surface $O_3$ formation sensitivity**

In addition to the significant diurnal variation, the surface $O_3$ formation sensitivity also has interannual variation. As discussed in Section 3.1, the $NO_2$ VCD in the HRB is decreasing at a rate of $0.07 \times 10^{15}$ mol/cm$^2$/year (-1.2% per year), while the HCHO VCD is increasing at a rate of $0.14 \times 10^{15}$ mol/cm$^2$/year (0.9% per year). Changes in $O_3$ precursor concentrations inevitably cause shifts in $O_3$ formation sensitivity, and we construct conventional FNR using TROPOMI observed $NO_2$ and

HCHO VCD from May to September 2018–2022. In order to avoid the misjudgment of $O_3$ formation sensitivity caused by arbitrary selection of FNR thresholds, A third–order polynomial model was applied to investigate the empirical relationship between TROPOMI FNR and surface $O_3$ VMRs, which has been widely used in other studies (Ren et al., 2022b; Jin et al., 2020; Wang et al., 2022). Since the TROPOMI observed surface $O_3$ VMRs can be obtained after November 2021 in China, we only collected the relationship between TROPOMI FNR and surface $O_3$ VMRs from May to September, 2022. The third–order

polynomial fitting relationship between surface $O_3$ VMRs and TROPOMI FNR is shown in Fig. 9a, assuming that the peak of the curve (with a slope of 0) marks the transition from the VOC–limited regime to the $NO_x$–limited regime, the transition regime is defined as a range of slopes between -3 and +3 (Ren et al., 2022b), as shaded in Fig.9a. Through the third–order polynomial model, the TROPOMI FNR threshold in HRB was determined, which are FNR <2.1 for VOC–limited regime, FNR>3.2 for $NO_x$–limited regime. The TROPOMI FNR thresholds are much higher than the ground–based MAX–DOAS

derived FNR. First of all, the effects of background and primary source HCHO were eliminated in ground–based MAX–DOAS observations. Secondly, due to the difference in the vertical distribution of HCHO and $NO_2$ concentrations. TROPOMI observes tropospheric VCD, and the ground–based MAX–DOAS measures surface concentrations, while $NO_2$ usually concentrates on the surface, and HCHO typically extends further up at higher concentrations in the atmosphere (usually reaches its maximum value at 100–200m) (Fig. S3).

TROPOMI observations showed that the surface $O_3$ formation sensitivity in the HRB had large spatial differences (Fig.S9), with VOC–limited regime and transition regime dominating in 2018 and 2022, and transition regime and $NO_x$–limited regime dominating in 2019–2021. In general, the FNR values in the HRB showed an upward trend, and the monthly mean values were





basically in transition regime and NO$_x$–limited regime (Fig.3c). Since the TROPOMI satellite usually transits around 13:30, the O$_3$ formation sensitivity is basically consistent with that of ground–based MAX–DOAS observations. The increase of FNR also leads to the change of the area proportion of O$_3$ formation sensitivity. Fig. 9b–d shows the trend of the area proportion of VOC–limited regime, transition regime, and NO$_x$–limited regime in the HRB, in which the area proportion of VOC–limited

regime and transition regime decreases at a rate of 0.81% and 0.32% per year, respectively. While the NO$_x$–limited regime area proportion increased at a rate of 1.13% per year. Due to China's strict control of NO$_x$ emissions in recent years, the surface O$_3$ formation sensitivity in many areas of China has shown a transition from the VOC–limited regime to the transition regime or NO$_x$–limited regime. In other words, the strict control of NO$_x$ emissions dominates the interannual variation of surface O$_3$ formation sensitivity. This phenomenon has been widely reported in previous studies. For example, for the summer surface O$_3$

formation sensitivity in the BTH and surrounding areas from 2011 to 2016, the proportion of the NO$_x$–limited regime areas increased from 38% to 65%, and the proportion of the VOC–limited regime areas decreased from 16% to 3% (Wu et al., 2018). the similar transition in surface O$_3$ formation sensitivity also occurs in Shanghai from 2010 to 2019, with the proportion of VOC–limited regime areas decreasing from 33.4% to 24.9%, and the proportion of transition regime areas increasing from 40.0% to 63.8% (Li et al., 2021a).

Overall, there are significant diurnal and interannual transition in surface O$_3$ formation sensitivity in the HRB, mainly attributed to diurnal variations in O$_3$ precursors and their unbalance emission reductions (significant reduction of NO$_x$ emissions while slightly increase in VOC emissions). The early morning (8:00–9:00) was mainly controlled by VOCs concentrations, which shifted to NO$_x$–limited regime by midday (12:00–14:00). In addition, the area proportion of VOC–limited regime was also declining, while the NO$_x$–limited regime area proportion was increasing. Therefore, due to the

substantial reduction of NO$_x$ emissions, the low (increased) and peak (decreased) surface O$_3$ concentrations showed diametrically opposite trends, and the surface O$_3$ formation sensitivity to VOCs is generally weakened year by year. Accordingly, the O$_3$ improvement benefits of VOCs emission reduction may become weaker, while the O$_3$ improvement benefits of NO$_x$ emission reduction become larger. Moreover, long–distance transport of VOCs has less impact on O$_3$ concentrations due to the chemical loss of VOCs caused by oxidation of OH radicals during transport, which makes the long–

distance transport of VOCs much weaker than that of NO$_x$ (Wang et al., 2023d). Therefore, NO$_x$ emission reduction also contributes to intercity and even long–distance NO$_x$ joint prevention and control to mitigate regional O$_3$ pollution.

### 3.5 Opposite trends of peak and low O$_3$ concentrations in the eastern China

The previous analyses showed that the trend of low O$_3$ concentrations in the HRB is increasing while the trend of peak O$_3$ concentrations is decreasing, and this interannual trend is mainly driven by anthropogenic emissions, that is, the effect of

substantial reductions in NO$_x$ emissions. In order to determine whether this variation characteristic is widespread in other cities in eastern China, the trends of low and peak O$_3$ concentrations by considering all cities in the densely populated areas in eastern China were investigated (105 cities in total) (Fig. 10).





In general, the trend of low $O_3$ concentrations in most cities in eastern China was increasing, and the trend of $2^{nd}$ $O_3$ percentile ranges from -0.2 to 0.5 ppb/year (-1.6–5.6% per year), while the trend of peak $O_3$ concentrations were decreasing, with a $98^{th}$ $O_3$ percentile trend between -1.1 and 0.2 ppb/year (-1.0–0.3% per year). The decrease trend of peak $O_3$ and the increase trend of low $O_3$ concentrations in the NCP are significantly greater than that in the MLYRP. The highest increase trend

in low $O_3$ concentrations was 0.5 ppb/year in Handan, and the lowest decrease trend in peak $O_3$ concentrations was 1.1 ppb/year in Baoding. On average (105 cities), the trends of the $2^{nd}$, $50^{th}$ and $98^{th}$ $O_3$ percentile in eastern China were -0.39 ppb/year (-0.5%), -0.02 ppb/year (-0.0%), and 0.06 ppb/year (0.3%), respectively (Fig.S10). The trend of $2^{nd}$, $50^{th}$ and $98^{th}$ $NO_2$ percentile in eastern China is -0.36 ppb/year (-1.8%), -0.22 ppb/year (-1.8%), 0.12 ppb/year (2.1%), respectively. The corresponding $O_3$ exceedance days (-1.1% per year) and exceedance hours (-1.6% per year) in eastern China are also decreasing. In addition, the

MLR model has also been used to separate the anthropogenic and meteorological components to the variation of urban surface $98^{th}$ and $2^{nd}$ $O_3$ percentiles concentrations in eastern China (Fig.S11). The influence of meteorological on the interannual trend of $98^{th}$ $O_3$ percentile trends is -0.033 ppb/year (11%), and it is mainly driven by anthropogenic components (-0.265 ppb/year, 89%). The interannual fluctuation of the mean $2^{nd}$ $O_3$ percentiles meteorological component (-0.020 ppb/year, -28%) was also smaller than that of the anthropogenic component (0.091 ppb/year, +128%). The RH at nighttime increased slowly in eastern

China during May–September 2017–2022 (Fig.S8b), the meteorological component has an inhibitory effect on the increase of $2^{nd}$ $O_3$ percentile trends. Thus, the $98^{th}$ and $2^{nd}$ $O_3$ percentile trends in eastern China mainly depend on anthropogenic components. Moreover, there is a significant positive correlation between the trend of mean $NO_2$ concentrations and the trend of the $98^{th}$ $O_3$ percentile during May–September 2017–2022 (Fig.11), with Pearson correlation R=0.42 (P<0.01). The trend of mean $NO_2$ concentrations during May–September 2017–2022 was roughly negatively correlated with the trend of the $2^{nd}$ $O_3$

percentiles, with Pearson correlation R=-0.23 (P=0.02). Thus, the significant reductions in $NO_x$ emissions have diametrically opposite effects on peak and low $O_3$ concentrations, and the decreased trend in low $O_3$ concentrations and increased trend in peak $O_3$ concentrations are widespread in urban agglomerations in eastern China.

Diurnal variations of $O_3$ formation sensitivity are not unique in the HRB. As early as the end of the last century, Kleinman et al. (Kleinman et al., 2005) found that the $O_3$ formation sensitivity in the United States shifted from VOC–limited regime in

the morning to $NO_x$–limited regime in the afternoon, which was caused by emission, photochemical aging, mixed layer height and air mass transport. Recently, Wang et al. (Wang et al., 2023a) combined the Global Ozone Monitoring Experiment (GOME-2B) and Ozone Monitoring Instrument (OMI) satellite observations to investigate the diurnal variation characteristics of $O_3$ formation sensitivity in BTH and its surrounding cities from May–September 2021, and found that the $O_3$ formation sensitivity was mainly in VOC–limited regime in the morning (~10:00), while in transition regime or $NO_x$–limited regime in the afternoon

(~14:00), suggesting that the transition of $O_3$ formation sensitivity from early morning VOC–limited regime to midday $NO_x$–limited regime is widespread in the urban agglomerations of the NCP. Li et al. (Li et al., 2013) also used model simulation to investigate that in most areas of the Pearl River Delta, the $O_3$ formation sensitivity in the morning is limited in VOC–limited regime, while in the $O_3$ peak period is in $NO_x$–limited regime. Moreover, the model results also show that reducing the $NO_x$ concentration in the Pearl River Delta will increase the mean $O_3$ concentration, but also depress the peak $O_3$ level. Therefore,



in response to the current severe $O_3$ pollution in China, it is necessary to take into account the regular transitions in $O_3$ formation sensitivity throughout the day when formulating $O_3$ prevention and control policies, when $O_3$ concentrations are about to peak, it is necessary to strictly control $NO_x$ emissions.

**3.6 Key meteorological and anthropogenic factors inducing ozone pollution**

The previous analysis has clarified the trend of background, typical, and peak $O_3$ trends in eastern China, and this section focuses on the key meteorological and anthropogenic factors triggering ozone pollution in the cities of eastern China. According to MLR results, the characteristics of RH, temperature, and wind (U and V components) in Hefei, Huaibei, and Tai' an during $O_3$ exceedance days and normal days are shown in Fig.12a–d. $O_3$ exceedances are usually characterized by high temperatures and low RH (Li et al., 2020a; Weng et al., 2022; Ding et al., 2023; Mousavinezhad et al., 2021). It is worth noting that there are certain differences in temperature and RH in different cities that are prone to $O_3$ exceedances. The average temperature and RH most prone to $O_3$ exceedance in Tai' an is generally 26.7°C and 61.8%, while those in Hefei and Huaibei are 29.3°C and 57.9%, 28.6°C and 54.4%, respectively. The influence of wind speed and direction on $O_3$ concentration is complicated. For Hefei, the southeasterly wind (u=-2.0 m/s, v=1.4 m/s) was most favorable for $O_3$ exceedance, which was mainly affected by the transmission of air pollutants from the YRD region, while for Huaibei and Tai' an, the weak southerly wind was most favorable for $O_3$ exceedance (v=0.7 m/s in Huaibei and v=1.0 m/s in Tai' an). The effects of meteorological conditions on $O_3$ pollution are relatively similar in each city, with some local differences. Therefore, it is necessary to establish some localized indicators to improve the accuracy of $O_3$ pollution forecasting and warning.

Overall, Meteorological conditions affect $O_3$ concentrations in three ways. The first is the effect on photochemical reaction rates (Bloomer et al., 2009), which are affected by rising temperatures and increasing solar radiation intensity, leading to higher $O_3$ concentrations. Secondly, the effect on $O_3$ precursors, high temperature promotes the increase of VOCs emission from land surface vegetation (Churkina et al., 2017), which further leads to the increase of $O_3$ concentration. While the favorable meteorological conditions, such as high wind speed and precipitation, can reduce the $O_3$ precursors concentrations (Mousavinezhad et al., 2021), thereby reducing $O_3$ concentration. Finally, the influence of transmission, regional transport affects the distribution of $O_3$ concentration in cities (Lang et al., 2021). However, the frequency of extreme weather is increasing (Jin et al., 2022; Zhang et al., 2015), such as super–heatwave event in 2022. In the context of the current warming climate, temperature can elevate $O_3$ concentrations by increasing photochemical rates and promoting natural emissions (e.g., soil emissions of $NO_x$, vegetation emissions of VOCs), and meteorological components may have an increasing influence on $O_3$ concentrations.

We further analyzed the characteristics of $O_3$ precursor concentrations and $O_3$ growth rates on $O_3$ exceedance and normal days. Compared with $O_3$ normal days, the HCHO concentrations in Hefei (5.0 vs 3.7 ppb), Huaibei (4.9 vs 3.5 ppb), and Tai' an (5.2 vs 3.3 ppb) on the $O_3$ exceedance days are substantially increased (Fig.12 e–f), while the difference in $NO_2$ concentrations is not significant, 4.6 vs 4.2 ppb in Hefei, 4.4 vs 4.8 ppb in Huaibei, and 4.8 vs 5.2 ppb in Tai' an. Satellite



observations also show similar characteristics (Fig.S12 and Fig.S13), and we investigated the differences in HCHO and $NO_2$ VCD between $O_3$ exceedance days and normal days for all cities in the HRB based on TROPOMI observations, and the HCHO concentrations in each city of HRB on $O_3$ exceedance days were higher than those on $O_3$ normal days, while the differences in $NO_2$ were not significant. Figure 12 g–h shows that the mean $O_3$ growth rate between 8:00 and 11:00 on $O_3$ exceedance days is significantly rapider than the mean $O_3$ growth rate on $O_3$ normal days. In contrast, the mean $O_3$ growth rate from 11:00 to 14:00 is only slightly faster than the mean $O_3$ growth rate on $O_3$ normal days. Most importantly, on $O_3$ exceedance days, the average $O_3$ growth rate between 8:00 and 11:00 is considerably faster than the average $O_3$ growth rate from 11:00 to 14:00, 13.3 vs 4.3 ppb in Hefei, 12.6 vs 6.6 ppb in Huaibei, and 12.8 vs 7.3 ppb in Tai'an. According to the analysis in Section 3.3, the $O_3$ formation sensitivity is usually in VOC–limited regime or transition regime from 8:00 to 11:00, while the $O_3$ formation sensitivity is usually in transition regime or $NO_x$–limited regime between 11:00 and 14:00. Therefore, the VOC concentrations play a key role in the rapid $O_3$ growth process, which can also explain why the HCHO concentrations increased significantly in all cities on $O_3$ exceedance days.

## 4 Conclusions

In this study, we investigated urban–scale $O_3$ trends in densely populated areas in eastern China using multi–source data. Through long–term records of surface $O_3$ and its related parameters at 105 urban air quality monitoring sites in eastern China from May–September 2017–2022, different trends were found for low, typical and peak surface $O_3$ concentrations under the current situation of uneven $O_3$ precursors emission reductions (significant reduction of $NO_x$ emissions while slightly increases in VOC emissions). The statistical results show that the low $O_3$ concentration in eastern China increases (0.06 ppb/year, 0.3% per year) significantly and the peak $O_3$ concentrations decreases (-0.39 ppb/year, -0.5% per year) considerably. The underlying cause of surface $O_3$ trends in eastern China is anthropogenic emissions, and the diametrically opposite trends in peak (decrease) and low (increase) $O_3$ concentrations were caused by the significant reductions in $NO_x$ emissions.

We also analyzed the diurnal and annual $O_3$ formation sensitivity through satellite and ground–based measurements. Based on the long–term MAX–DOAS observations in Hefei, Huaibei and Tai 'an, the surface $O_3$ formation sensitivity in early morning (8:00–9:00) was mainly controlled by VOCs concentrations, which shifted to $NO_x$–limited regime by midday (12:00–14:00). Moreover, $O_3$ formation sensitivity is in VOC–limited regime or transition regime during periods (8:00–11:00) of sharp increases in $O_3$ concentrations, and it is usually in transition regime or $NO_x$–limited regime when the $O_3$ concentration reaches its peak (11:00–14:00). Therefore, reducing the VOCs concentrations is the key to reverse the current high $O_3$ level situations, and in order to further depress the peak $O_3$ concentrations and reduce the number of $O_3$ exceedance days, the control of $NO_x$ concentrations should not be neglected. In addition, there are significant interannual transition in surface $O_3$ formation sensitivity, the area proportion of VOC–limited regime and transition regime in HRB decreases at a rate of 0.81% and 0.32% per year, respectively. While the $NO_x$–limited regime area proportion in HRB increased at a rate of 1.13% per year, which is also attributed to unbalanced $O_3$ precursor emission reductions.



Our results highlight the positive impact of $NO_x$ reduction in controlling peak $O_3$ levels. It also points out the centrality of reducing VOCs concentrations in the current $O_3$ pollution prevention and control in eastern China. In response to the current severe $O_3$ pollution in China, the regular changes in $O_3$ formation sensitivity throughout the day should be taken into account when formulating $O_3$ prevention and control policies. This study providers novel insight into spatiotemporal variability in

ozone formation sensitivity in eastern China, and will be further expanded at different altitude levels in our future studies.

*Author contributions.* Cheng Liu, Chengzhi Xing, and Yujia, Chen conceived and supervised the study; Zhuang Wang analyzed the data; Zhuang Wang wrote the manuscript with input from Cheng Liu, Chengzhi Xing, and Yujia Chen; Chune Shi, and Hao Zhang reviewed and commented on the paper; All authors contributed to discuss the results and revised manuscript.

*Competing interests.* The authors declare that they have no competing interests.

*Acknowledgments.* We would like to express our gratitude to Fusheng Mou and Wensu Li of Huaibei Normal University for their assistance. We would like to thank the Ministry of Ecology and Environment of China for the $O_3$, $NO_2$ and CO data. We would like to thank the ERA5 data developers and TROPOMI data developers for providing free and open source materials. This research was supported by grants from the Anhui Provincial Natural Science Foundation "Jianghuai Meteorological" Joint

Fund (2208085UQ04), National Natural Science Foundation of China (U21A2027, 42207113), Anhui Meteorological Bureau Special Programme for Innovation and Development (CXB202303), and East China Regional Meteorological Science and Technology Collaborative Innovation Fund (QYHZ202317).

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



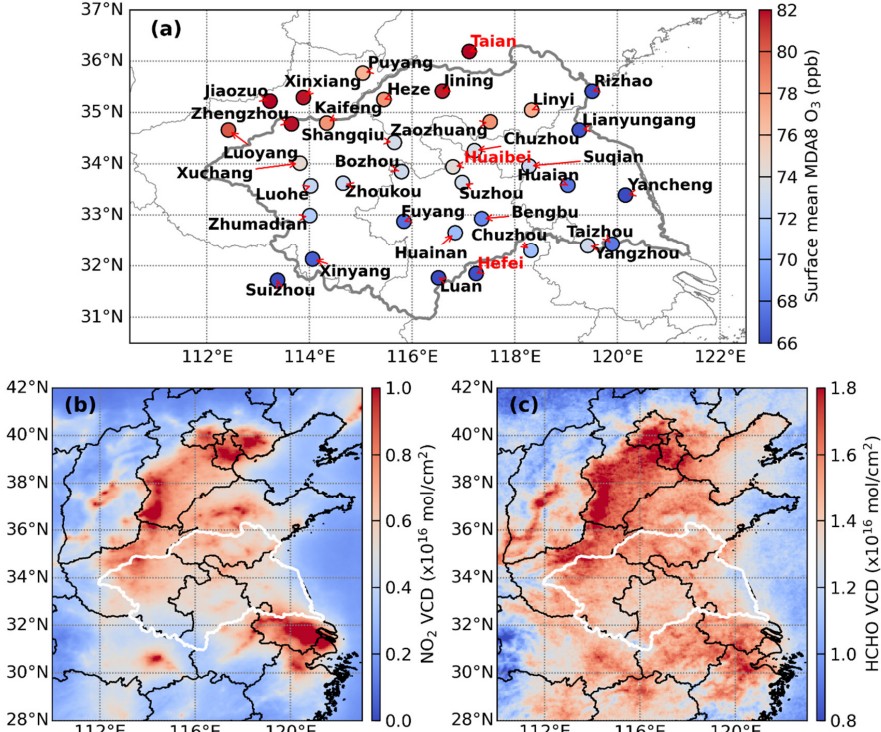

**Fig.1**. Spatial distributions of O₃ and its precursors in the HRB. (a) Spatial distributions of surface mean MDA8 O₃ concentrations during May–September 2017–2022. The red arrow indicates the name of each city, and the city name marked in red are equipped with ground–based MAX–DOAS observations. Spatial distributions of tropospheric mean (b) NO₂ and (c) HCHO VCD during May–September 2018–2022.



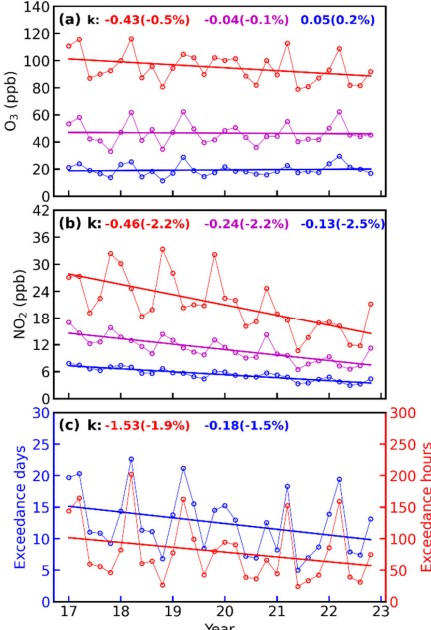

**Fig.2**. Trends of surface (a) $O_3$, (b) $NO_2$, (c) $O_3$ exceedance days and $O_3$ exceedance hours in HRB during May–September 2017–2022. The red, magenta, and blue solid lines in (a) and (b) indicate the trends for the 98th, 50th, and 2nd percentiles, respectively. The labels on (a) and (b) represent the trends in $O_3$ and $NO_2$ for May–September 2017–2022, units: ppb/year. The labels on (c) represent the trends in $O_3$ exceedance days and $O_3$ exceedance hours for May–September 2017–2022. The percentage change is indicated in brackets.




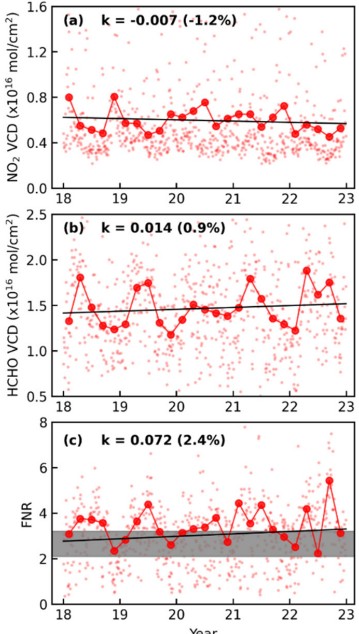

**Fig.3**. Trends of TROPOMI observed (a) NO$_2$ VCD, (b) HCHO VCD, and (c) FNR values averaged over HRB during May–September 2018–2022. The labels at the top of each panel represent the trends in NO$_2$ VCD, HCHO VCD, and FNR, respectively. The percentage change is indicated in brackets. The light red dots in (a–c) represent the daily values, and the solid red dots are monthly mean values. The horizontal shadow in (c) represents the transition regime, the top of the shadow represents the NO$_x$–limited regime, and the bottom of the shadow represents the VOC–limited regime.

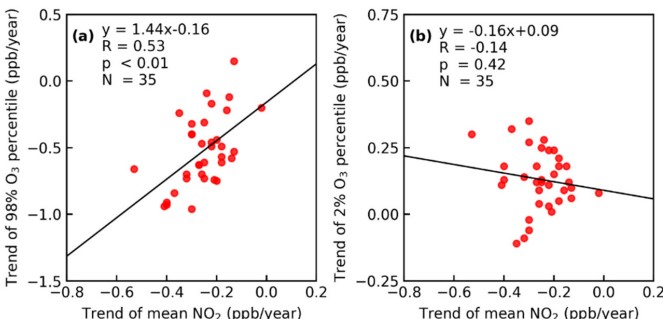

**Fig.4**. Scatterplots showing the relationships between the (a) trend of mean NO$_2$ concentrations and the trend of 98[th] O$_3$ percentiles, (b) trend of NO$_2$ concentrations and trend of 2[nd] O$_3$ percentiles in each city of HRB during May–September 2017–2022. The correlation coefficients are shown in the top left of each panel, N=number of cities.



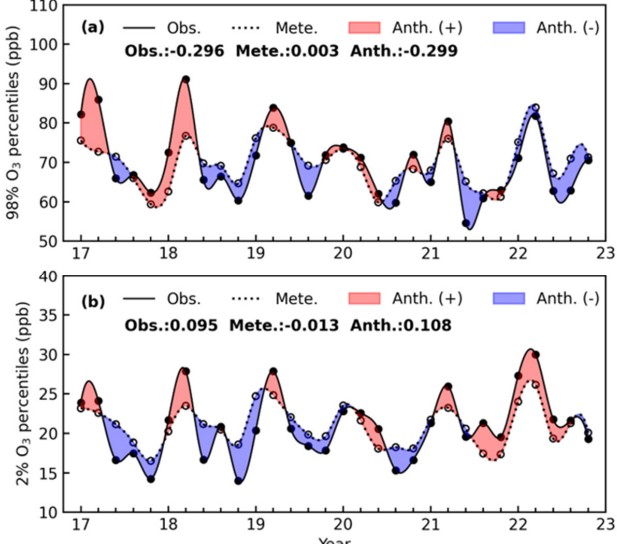

**Fig.5**. Variations in observed (a) 98th and (b) 2nd $O_3$ percentiles (solid lines connected by solid black dots), meteorological (a) 98th and (b) 2nd $O_3$ percentiles component (dotted lines connect black hollow points) in MLR simulations, and the anthropogenic (a) 98th and (b) 2nd $O_3$ percentiles component (red and blue shading) in HRB during May–September 2017–2022. The labels at the top of each panel represent the trend in observed, meteorological, and anthropogenic components.

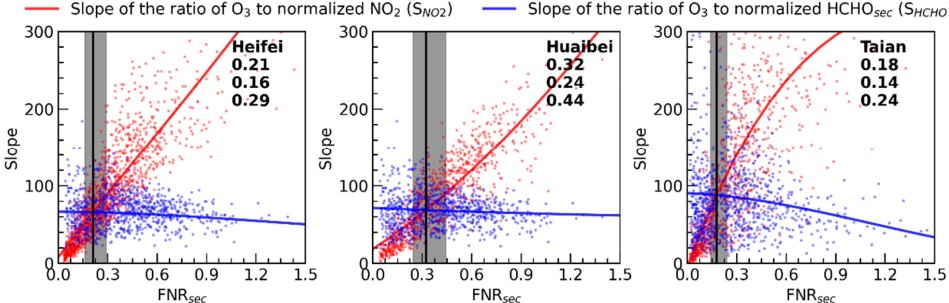

**Fig.6**. Three–order fitting of slopes of $O_3$ VMRs versus normalized $NO_2$ VMRs and slopes of $O_3$ VMRs versus normalized secondary HCHO VMRs in different $FNR_{sec}$ values in Hefei, Huaibei, and Tai'an during May–September based on MAX–DOAS observations. The intersect at $FNR_{sec}$ indicated by the black solid line. The vertical shadow indicates the relative difference between the slopes of $O_3$ VMRs versus normalized $NO_2$ VMRs and slopes of $O_3$ VMRs versus secondary HCHO VMRs within 25% (transition regime). The labels at the top right of each panel represent the intersect $FNR_{sec}$ values and the thresholds for the $NO_x$–limited regime (high) and VOC–limited regime (low) in Hefei, Huaibei, and Tai'an, respectively.



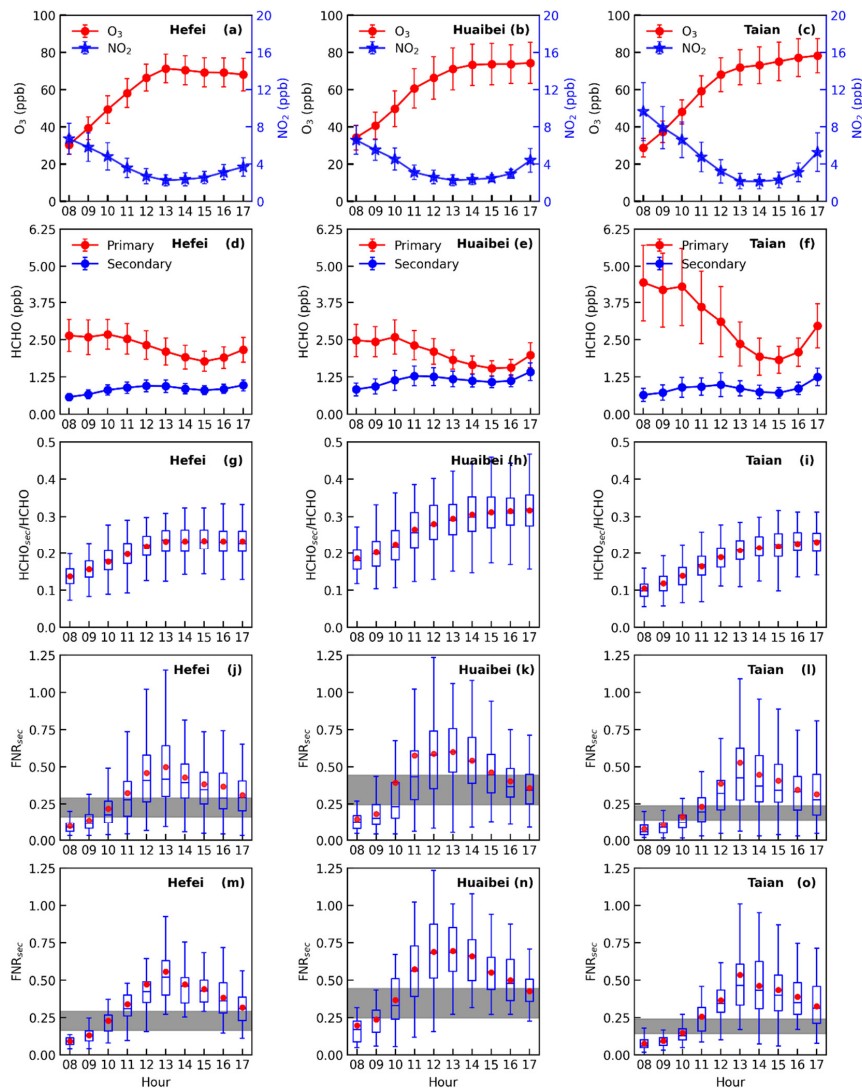

**Fig.7**. Diurnal variation of surface (a–c) $O_3$ and $NO_2$ VMRs, (d–f) HCHO VMRs contributed by primary and secondary sources, (g–i) the ratio of secondary HCHO to total HCHO VMRs, (j–l) $FNR_{sec}$ during the whole observation, and (m–o) $FNR_{sec}$ during $O_3$ exceedance day in Hefei, Huaibei, and Tai'an during May–September, respectively. The vertical bars in (a–f) represent the one standard deviation. The dot within the box indicates the mean value, the positions of box plots represent the 5th, 25th, 50th, 75th, 95th percentiles, respectively. The horizontal shadow in (j–o) represents the transition regime, the top of the shadow represents the $NO_x$–limited regime, and the bottom of the shadow represents the VOC–limited regime.





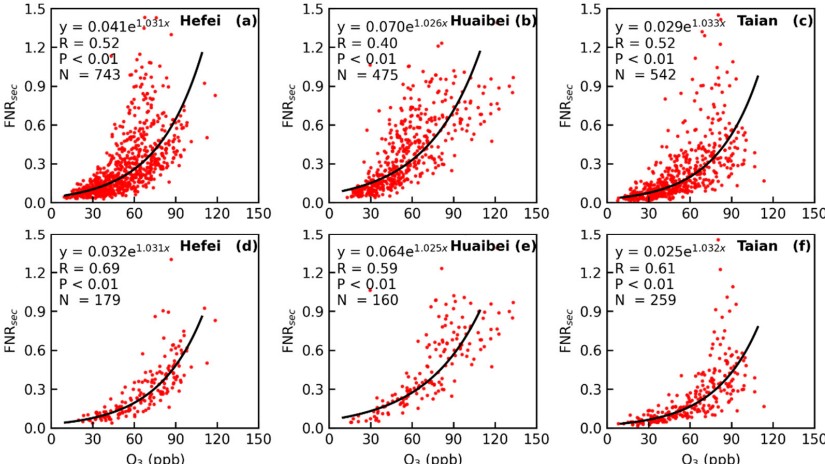

**Fig.8**. The relationship between the $O_3$ VMRs and $FNR_{sec}$ during (a–c) the whole observation, and (d–f) the $O_3$ exceedance day from 08:00 to 13:00 in Heifei, Huaibei, and Tai'an, respectively. The black line represents the exponential fitting ($f(x) = A \times e^{B \times x}$). The fitting functions and correlation coefficients for exponential fit are shown at the top left of each panel, N=number of samples.



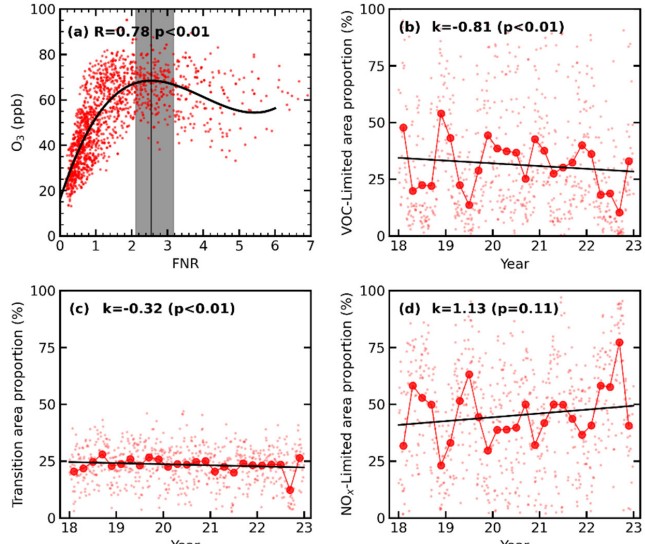

**Fig.9.** (a) Variation of monthly mean $O_3$ VMRs (~13:30) with monthly mean TROPOMI FNR in HRB during May–September 2022. The solid line represents third–order polynomial fitting. The vertical line represents the maximum value of the fitted curve, and the vertical shadow represents the range of the curve slope from −3 to +3 (transition regime). Trends of TROPOMI observed area proportion for (b) VOC–limited regime, (c) Transition regime, and (c) $NO_x$–limited regime over HRB during May–September 2018–2022. The light red dots in (b–d) represent the daily values, and the solid red dots are monthly mean values.



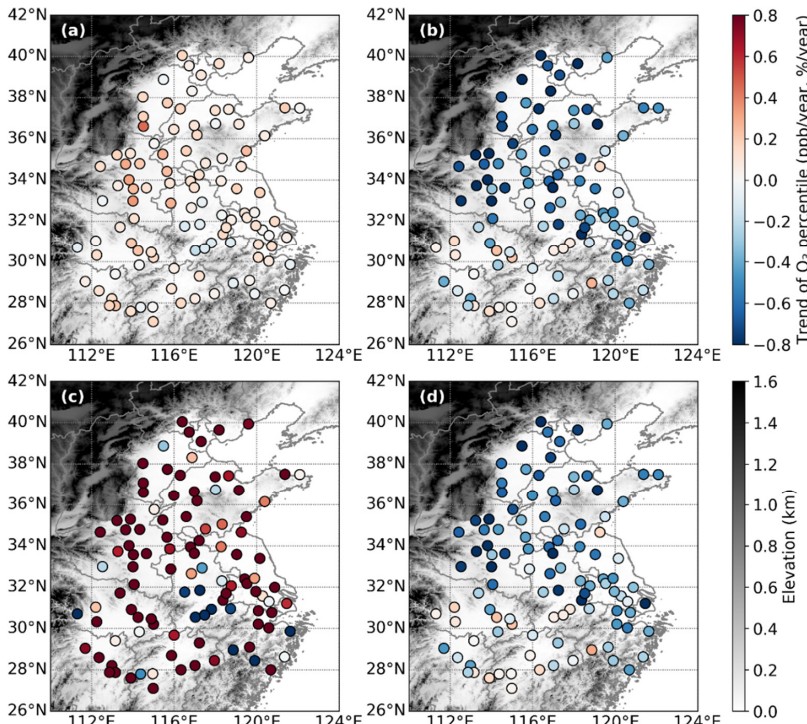

**Fig.10**. Trends of peak and low $O_3$ concentrations in eastern China. Trend of (a) 2% and (b) 98% $O_3$ percentiles, units: ppb/year, percentage variations in (c) 2% and (d) 98% $O_3$ percentile, units: %/year.

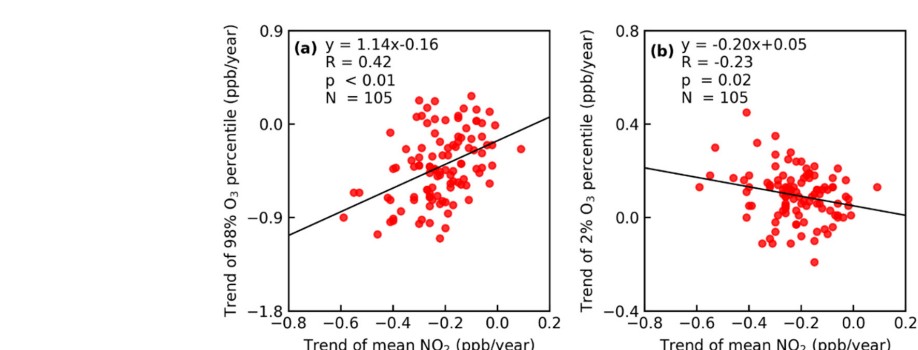

**Fig.11**. The same as Fig.4 but in each city of eastern China.



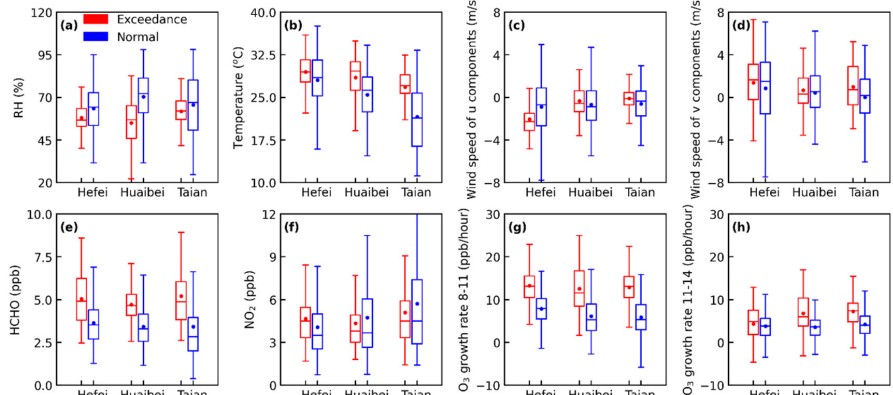

**Fig.12**. Box plot of (a) RH, (b) temperature, (c) wind speed of u components, (d) wind speed of v components, (e) HCHO VMRs, (f) $NO_2$ VMRs, (g) $O_3$ growth rate between 8:00 and 11:00, and (h) $O_3$ growth rate between 11:00 and 14:00 in Hefei, Huaibei, and Tai'an during $O_3$ exceedance days and normal days. The dot within the box indicates the mean value, the positions of box plots represent the 5th, 25th, 50th, 75th, 95th percentiles, respectively.

