# Peer review of "Opposing trends in the peak and low ozone concentrations in eastern"

_EGUsphere, 2024_

## Author Comment (AC1)

**Response to Reviewer 1 Comments**

We truly grateful for the reviewers for the valuable and constructive comments, which are very useful for the improvement of the manuscript. We have revised the manuscript carefully according to the reviewers' comments. Point–to–point responses are given below. The original comments are black in color, while our responses are in blue. The revised parts in the manuscript are marked in red. All the page number and line number are referred to the revised manuscript.

**General comments:**

**Point 1:** It's hard for me to identify the main topic of this paper. While the primary focus appears to be explaining the opposing trends in the 98$^{th}$ and 2$^{nd}$ percentiles of ozone in the MLYRP (part of eastern China), the paper includes many other analyses that are irrelevant to this topic. For instance, Section 3.4 (Interannual differences in surface $O_3$ formation sensitivity) and Section 3.6 (Key meteorological and anthropogenic factors inducing $O_3$ pollution) should be removed (they haven't found anything new in these sections either). The entire paper should be reframed. Additionally, Section 3.5 presents the same analysis and draws the same conclusions as Sections 3.1–3.3 but for a larger area (eastern China). The inclusion of Section 3.5 seems redundant. Why don't the authors focus on eastern China throughout the manuscript?

**Response 1:** We apologize for the confusion generated by the previous version of the manuscript and sincerely hope that our logic is now easier to follow with this new version. We have followed the reviewer's comments and reorganized the manuscript to avoid diverting the reader's attention. Firstly, we have deleted sections 3.4 (Interannual differences in surface $O_3$ formation sensitivity) and 3.6 (Key meteorological and anthropogenic factors inducing $O_3$ pollution) in the previous manuscript. Then, sections 3.1–3.3 and 3.5 was reframed, and the entire updated manuscript focuses on eastern China throughout the manuscript.

The new manuscript mainly includes four parts. First, we report long–term records of surface $O_3$ and related parameters observed at urban air quality monitoring sites and by satellites in eastern China, characterizing the trends of low, typical, and peak surface $O_3$ concentrations during the warm season (May–September) from 2017 to 2022. Then, a Multiple Linear Regression (MLR) model is used to evaluate the anthropogenic and meteorological contributions to the 98$^{th}$ and 2$^{nd}$ $O_3$ percentile trends. Next, secondary formaldehyde (HCHO) and $NO_2$ are employed to diagnose the diurnal variations in $O_3$ formation sensitivity and investigate the reasons for peak $O_3$ concentration trends in the context of current $NO_x$ reduction.

Finally, we discuss the reasons for the potential increase in low O$_3$ concentrations and the sensitivity of peak and low O$_3$ trends during the study period. Please refer to our new manuscript for details.

In addition, following the referee's suggestion, the manuscript was edited by Elsevier Language Editing Services (please see the following Elsevier certificate).

[Figure]

**Fig. R1** Certificate of Elsevier language editing services

**Point 2:** To understand the drivers of ozone trends, the authors use Multiple Linear Regression (MLR) to separate meteorological influences. There are several issues with their analysis: (a) The predicted variable, O$_3$ concentration, should also be normalized in the MLR. Normalizing Y first would eliminate the need for natural background O$_3$ in equation (2). (b) Line 27, Page 6, the statement that O$_3$ from natural sources is stable is incorrect. Biogenic VOCs and soil NOx emissions are highly sensitive to temperature. (c) Lines 19–21, 23–24, Page 10, the authors seem to confuse the terms "interannual fluctuation" and "trend." The trend observed is actually a low–frequency signal after removing the high–frequency signals (interannual fluctuation). In this part, it is only acceptable to conclude that the anthropogenic component drives the trend, but it is apparent that meteorological parameters dominate the interannual fluctuation (as it roughly reproduces the peaks and troughs). All of this needs to be corrected.

**Response 2:** We thank the reviewer for pointing out this issue. (a) We corrected the problem (the predicted variable, O$_3$ concentration, should also be normalized in the MLR. Normalizing Y first would eliminate the need for natural background O$_3$ in equation 2) in the MLR model, and rearranged this subsection to conclude that, although anthropogenic emissions are the main driver of the opposing trends in peak and low O$_3$ concentrations in eastern China, the effect of

the change of meteorological component on $2^{nd}$ or $98^{th}$ $O_3$ percentiles trends cannot be ignored. (b) The statements in Line 27, Page 6 have been deleted, and (c) Lines 19–21, 23–24, Page 10 have been corrected.

In the new manuscript, we used the same stepwise MLR modeling approach as (Zhai et al., 2019; Li et al., 2018; Li et al., 2020; Liu et al., 2023). Following Liu et al. (2023) and Li et al. (2020), the MLR model fitted the deseasonalized and detrended 10 d mean $98^{th}$ or $2^{nd}$ $O_3$ percentile time series to the deseasonalized and detrended 10 d mean meteorological variables. The deseasonalized and detrended time series data were constructed by removing the 50 d moving average data from the 10 d moving average data. The stepwise MLR model has the following form:

$$Y(t) = R + \sum_{k=1}^{n} \beta_k X_k(t) \tag{1}$$

Where $Y(t)$ is the deseasonalized and detrended daily surface $98^{th}$ or $2^{nd}$ $O_3$ percentile time series, R is the regression constant, $\beta_k$ is the regression coefficient, and $X_k$ is the deseasonalized and detrended daily meteorological variable considered as a possible $O_3$ covariate. Stepwise regressions were performed, adding and removing terms based on their independent statistical significance to obtain the best model fit.

Daily meteorological variables were obtained from the ERA5 reanalysis data (Download from https://cds.climate.copernicus.eu, last access: January 7, 2024), included temperature (T, °C), surface relative humidity (RH, %), total cloud cover (TCC), total precipitation (TP, mm), mean sea level pressure (MSLP, hPa), wind speed of U, V components (U, V, m/s), boundary layer height (BLH, m), and vertical velocity at 850 hPa (V850, m/s).

First, we used the MLR model to remove the effects of meteorological variability from the 2017 to 2022 $98^{th}$ or $2^{nd}$ $O_3$ percentile trends. We apply Eq. (1) to the meteorological anomalies $X_k$ during May–September 2017–2022, obtained by removing the 6–year means of the 50 d moving averages from the 10 d mean time series. The anomalies calculated in this process were deseasonalized but not detrended. This yields the meteorology–driven $98^{th}$ or $2^{nd}$ $O_3$ percentile anomalies $Y_m(t)$

$$Y_m(t) = R + \sum_{k=1}^{n} \beta_k X_k(t) \tag{2}$$

Secondly, to avoid overfitting, only the three most important meteorological parameters were selected based on their individual contributions to the regressed $98^{th}$ or $2^{nd}$ $O_3$ percentiles, along with the requirement that they be statistically significant above the 95% confidence level

in the MLR model (Li et al., 2018). The fit results and selected meteorological variables varied by city but were regionally consistent (Table R1 and Table R2). The 98th or 2nd O$_3$ percentile anomalies $Y_a(t)$ obtained by deseasonalizing, but not detrending, the 98th or 2nd O$_3$ percentile time series in a similar manner as for the meteorological variables (by removing the 6–year means of the 50 d moving averages). The residual anomaly $Y_r(t)$ after removing the meteorology–driven 98th or 2nd O$_3$ percentile anomalies from the MLR model is given by

$$Y_r(t) = Y_a(t) - Y_m(t) \tag{3}$$

Finally, the residual is an anomalous component that cannot be explained by the MLR meteorological model and is referred to as meteorologically corrected data by (Zhai et al., 2019). It consists of noise due to the limitations of the MLR model and other factors and can be mainly attributed to long–term trends in anthropogenic emission changes over a 6–year period. The trend in the regressed 98th or 2nd O$_3$ percentile reflected the meteorological contribution, and the residual was then used to reflect the presumed anthropogenic contribution. For the updated MLR model, please refer to section 2.4 in the manuscript.

**Table R1.** Meteorological drivers of 2% O$_3$ percentile and Pearson correlation coefficient between observed and modeled 2% O$_3$ percentile in each city of eastern China during May–September 2017–2022

| | Meteorological variable | | | R | | Meteorological variable | | | R |
|---|---|---|---|---|---|---|---|---|---|
| | 1st | 2st | 3st | | | 1st | 2st | 3st | |
| Taian | T | RH | U | 0.35 | Beijing | U | RH | T | 0.23 |
| Puyang | T | BLH | RH | 0.50 | Tianjing | U | T | RH | 0.32 |
| Rizhao | U | V | T | 0.46 | Baoding | U | RH | TP | 0.27 |
| Jining | RH | T | V | 0.54 | Lanfang | U | T | V | 0.23 |
| Xinxiang | U | RH | V | 0.41 | Shijiazhuang | RH | BLH | U | 0.27 |
| Jiaozuo | T | RH | U | 0.39 | Handan | RH | V | BLH | 0.20 |
| Heze | RH | T | V | 0.47 | Qinghuangdao | V | U | RH | 0.32 |
| Linyi | RH | U | TP | 0.40 | Cangzhou | BLH | T | MSLP | 0.28 |
| Kaifeng | RH | U | T | 0.49 | Xingtai | RH | BLH | U | 0.17 |
| Zhengzhou | BLH | RH | U | 0.47 | Hengshui | T | V | RH | 0.28 |
| Luoyang | BLH | RH | V | 0.16 | Tangshan | U | V | RH | 0.20 |
| Zaozhuang | RH | T | U | 0.46 | Jinan | V | BLH | RH | 0.59 |
| Lianyungang | RH | V850 | U | 0.37 | Qingdao | V | BLH | RH | 0.26 |
| Shangqiu | RH | T | V | 0.48 | Zibo | V | BLH | RH | 0.59 |
| Xuzhou | RH | BLH | V | 0.48 | Dongying | U | V | V850 | 0.36 |
| Xuchang | BLH | RH | U | 0.53 | Yantai | V | BLH | U | 0.33 |
| Suqian | RH | T | MSLP | 0.40 | Weifang | BLH | T | U | 0.34 |
| Huaibei | RH | U | BLH | 0.59 | Weihai | RH | U | MSLP | 0.36 |
| Pingdingshan | BLH | RH | U | 0.57 | Dezhou | RH | V | BLH | 0.40 |
| Bozhou | RH | V | T | 0.49 | Liaocheng | BLH | V | RH | 0.48 |
| Zhoukou | RH | U | T | 0.49 | Binzhou | BLH | T | V | 0.23 |
| Luohe | RH | U | MSLP | 0.45 | Shaoxing | RH | BLH | T | 0.59 |
| Suzhou | RH | U | BLH | 0.50 | Jinhua | RH | TP | V | 0.60 |
| Huaian | RH | V | TP | 0.42 | Taizhou | V | RH | U | 0.54 |
| Yancheng | V | RH | BLH | 0.37 | Ningbo | RH | V | TP | 0.48 |

| | | | | | | | | | |
|---|---|---|---|---|---|---|---|---|---|
| Nanyang | RH | U | BLH | 0.67 | Wuhan | RH | V | BLH | 0.61 |
| Zhumadian | RH | TP | BLH | 0.52 | Changsha | RH | V | BLH | 0.71 |
| Fuyang | RH | V | U | 0.64 | Jinghzou | RH | V | BLH | 0.55 |
| Bengbu | RH | BLH | U | 0.36 | Yueyang | RH | BLH | V | 0.60 |
| Huainan | RH | BLH | U | 0.56 | Zhuzhou | RH | V | BLH | 0.73 |
| Xinyang | RH | TP | T | 0.63 | Xiangtan | RH | V | BLH | 0.72 |
| Suizhou | RH | BLH | U | 0.63 | Yichang | RH | U | V850 | 0.64 |
| Shanghai | RH | V | TP | 0.55 | Yiyang | RH | V | BLH | 0.67 |
| Nanjing | V | RH | T | 0.40 | Changde | V | BLH | RH | 0.59 |
| Wuxi | RH | V | BLH | 0.57 | Jingmen | V | RH | U | 0.63 |
| Changzhou | V | RH | TP | 0.49 | Huangshi | RH | U | V850 | 0.72 |
| Suzhou | RH | V | T | 0.52 | Huanggang | RH | U | V850 | 0.77 |
| Nantong | V | RH | U | 0.62 | Xianning | RH | U | MSLP | 0.74 |
| Yangzhou | V | RH | T | 0.52 | Xiaogan | RH | V850 | BLH | 0.68 |
| Zhenjiang | V | RH | T | 0.55 | Quzhou | V | RH | U | 0.68 |
| Taizhou | V | RH | T | 0.59 | Lishui | RH | V | T | 0.70 |
| Luan | RH | V | T | 0.24 | Wenzhou | RH | V | U | 0.63 |
| Hangzhou | RH | V | BLH | 0.53 | Jiujiang | V | RH | MSLP | 0.58 |
| Jiaxing | RH | V | T | 0.45 | Nanchang | V | RH | V850 | 0.67 |
| Huzhou | V | RH | U | 0.61 | Jingdezhen | BLH | V850 | T | 0.59 |
| Hefei | RH | TP | V | 0.37 | Shangrao | V850 | BLH | RH | 0.65 |
| Wuhu | V | TP | U | 0.46 | Yingtan | BLH | V850 | RH | 0.64 |
| Maanshan | V | U | TP | 0.45 | Yichun | V850 | U | T | 0.63 |
| Tonglin | V | T | TP | 0.50 | Fuzhou | RH | V | BLH | 0.72 |
| Anqing | V | T | TP | 0.56 | Jian | BLH | U | RH | 0.57 |
| Chuzhou | RH | V | BLH | 0.44 | Xinyu | BLH | V | U | 0.62 |
| Chizhou | RH | V | BLH | 0.47 | Pingxiang | V850 | BLH | T | 0.61 |
| Xuancheng | T | RH | U | 0.27 | - | - | - | - | - |

**Table R2.** Meteorological drivers of 98% $O_3$ percentile and Pearson correlation coefficient between observed and modeled 98% $O_3$ percentile in each city of eastern China during May–September 2017–2022

| | Meteorological variable | | | | | Meteorological variable | | | |
|---|---|---|---|---|---|---|---|---|---|
| | 1st | 2st | 3st | R | | 1st | 2st | 3st | R |
| Taian | T | TP | TCC | 0.71 | Beijing | MSLP | BLH | V | 0.30 |
| Puyang | T | V850 | TP | 0.79 | Tianjing | V | MSLP | BLH | 0.27 |
| Rizhao | T | U | TCC | 0.67 | Baoding | MSLP | V | BLH | 0.28 |
| Jining | T | TP | V850 | 0.77 | Lanfang | MSLP | BLH | V | 0.32 |
| Xinxiang | T | TCC | BLH | 0.74 | Shijiazhuang | MSLP | BLH | V | 0.24 |
| Jiaozuo | T | TP | TCC | 0.81 | Handan | T | MSLP | U | 0.31 |
| Heze | T | V850 | TP | 0.78 | Qinghuangdao | TCC | U | MSLP | 0.36 |
| Linyi | T | TP | TCC | 0.73 | Cangzhou | V | BLH | MSLP | 0.35 |
| Kaifeng | T | V850 | TP | 0.78 | Xingtai | MSLP | BLH | V | 0.25 |
| Zhengzhou | T | V850 | TP | 0.78 | Hengshui | V | MSLP | BLH | 0.38 |
| Luoyang | U | TCC | V | 0.28 | Tangshan | TCC | MSLP | T | 0.27 |
| Zaozhuang | T | TP | TCC | 0.81 | Jinan | T | TP | V850 | 0.77 |
| Lianyungang | TCC | U | TP | 0.68 | Qingdao | U | BLH | RH | 0.45 |
| Shangqiu | T | TP | V850 | 0.71 | Zibo | T | TP | V850 | 0.70 |
| Xuzhou | T | TCC | TP | 0.74 | Dongying | T | RH | V | 0.66 |
| Xuchang | T | V850 | TCC | 0.70 | Yantai | U | T | RH | 0.44 |
| Suqian | RH | TCC | TP | 0.74 | Weifang | T | TP | U | 0.68 |
| Huaibei | TCC | T | TP | 0.74 | Weihai | U | T | RH | 0.44 |
| Pingdingshan | T | V850 | U | 0.66 | Dezhou | T | TP | V850 | 0.75 |
| Bozhou | TCC | T | TP | 0.67 | Liaocheng | T | V850 | TP | 0.77 |
| Zhoukou | T | TCC | RH | 0.73 | Binzhou | T | TP | V | 0.66 |
| Luohe | T | V850 | RH | 0.75 | Shaoxing | T | V | TP | 0.66 |

| | | | | | | | | | |
|---|---|---|---|---|---|---|---|---|---|
| Suzhou | TCC | T | TP | 0.71 | Jinhua | T | V | TP | 0.59 |
| Huaian | T | TCC | TP | 0.72 | Taizhou | U | V | T | 0.62 |
| Yancheng | TP | V850 | U | 0.56 | Ningbo | T | V | U | 0.69 |
| Nanyang | T | TCC | RH | 0.74 | Wuhan | TCC | RH | V850 | 0.75 |
| Zhumadian | TCC | BLH | TP | 0.67 | Changsha | RH | TCC | V | 0.76 |
| Fuyang | RH | TCC | MSLP | 0.74 | Jinghzou | RH | V850 | T | 0.84 |
| Bengbu | T | TCC | TP | 0.72 | Yueyang | RH | TCC | U | 0.82 |
| Huainan | TCC | RH | MSLP | 0.68 | Zhuzhou | RH | V | T | 0.73 |
| Xinyang | RH | TCC | U | 0.74 | Xiangtan | RH | V | TCC | 0.78 |
| Suizhou | RH | TCC | MSLP | 0.75 | Yichang | RH | TCC | V850 | 0.79 |
| Shanghai | T | U | RH | 0.72 | Yiyang | RH | TCC | V850 | 0.80 |
| Nanjing | TCC | V850 | V | 0.61 | Changde | TCC | V850 | T | 0.74 |
| Wuxi | TP | TCC | U | 0.66 | Jingmen | RH | V850 | T | 0.76 |
| Changzhou | TCC | TP | BLH | 0.60 | Huangshi | RH | TCC | V | 0.76 |
| Suzhou | T | TP | U | 0.63 | Huanggang | TCC | RH | U | 0.72 |
| Nantong | T | RH | BLH | 0.75 | Xianning | RH | V850 | T | 0.80 |
| Yangzhou | TCC | TP | V850 | 0.61 | Xiaogan | RH | TCC | U | 0.72 |
| Zhenjiang | TCC | TP | MSLP | 0.61 | Quzhou | RH | V850 | T | 0.74 |
| Taizhou | TCC | V850 | U | 0.61 | Lishui | RH | U | V | 0.65 |
| Luan | TCC | MSLP | T | 0.61 | Wenzhou | U | RH | V | 0.65 |
| Hangzhou | TP | TCC | T | 0.69 | Jiujiang | TCC | V850 | RH | 0.80 |
| Jiaxing | T | TP | U | 0.70 | Nanchang | RH | V850 | T | 0.72 |
| Huzhou | TP | TCC | T | 0.67 | Jingdezhen | RH | V850 | TCC | 0.75 |
| Hefei | TCC | TP | V | 0.59 | Shangrao | TCC | V850 | T | 0.76 |
| Wuhu | TCC | TP | V | 0.65 | Yingtan | TCC | V850 | T | 0.78 |
| Maanshan | TCC | TP | MSLP | 0.66 | Yichun | V850 | TCC | T | 0.75 |
| Tonglin | TCC | MSLP | V | 0.63 | Fuzhou | V850 | T | TCC | 0.82 |
| Anqing | TCC | V | TP | 0.58 | Jian | TCC | T | V | 0.66 |
| Chuzhou | TCC | TP | MSLP | 0.58 | Xinyu | V850 | TCC | V | 0.75 |
| Chizhou | TCC | V | MSLP | 0.63 | Pingxiang | V850 | TCC | V | 0.75 |
| Xuancheng | TCC | V | MSLP | 0.58 | - | - | - | - | - |

**Point 3:** The authors conclude that the continuous NO$_x$ reduction during 2017–2022 is the reason for the differences in tendencies in the O$_3$ 98$^{th}$ and 2$^{nd}$ percentile trends. However, if you look carefully at Figure 5, which presents the anthropogenic impact for each year on the 98$^{th}$ and 2$^{nd}$ percentile trends, respectively, you will find that the anthropogenic impact shows a similar pattern for both trends until mid–2021. It appears that something that occurred after 2021 is the main reason for the divergence. More investigation is clearly needed. Also, how is this opposing trend sensitive to the period studied?

**Response 3:** Thanks for your constructive comments! We have corrected the problem in the MLR model according to Point 2, and the updated results is shown in Fig.R2. The trends for the monthly mean observed, meteorological, and anthropogenic 98$^{th}$ O$_3$ percentiles concentrations during May–September 2017–2021 are -0.363 ppb/year, -0.119 ppb/year (-33%), and -0.244 ppb/year (-67%), respectively (Table R3), and the trends for the monthly mean observed, meteorological, and anthropogenic 2$^{nd}$ O$_3$ percentiles concentrations are 0.027 ppb/year, -0.044 ppb/year (-163%), 0.071 ppb/year (263%), respectively. However, the trends

of monthly mean observed, meteorological and anthropogenic of 98$^{th}$ O$_3$ percentiles during May–September 2017–2022 are -0.178 ppb/year, 0.005 ppb/year (3%) and -0.183 ppb/year (-103%), respectively, and the trends of the observed, meteorological and anthropogenic of 2$^{nd}$ O$_3$ percentiles during May–September 2017–2022 are 0.115 ppb/year, 0.008 ppb/year (7%) and 0.107 ppb/year (93%), respectively.

Although anthropogenic emissions dominated variations in O$_3$ trends (May–September 2017–2022 and May–September 2017–2021), meteorological effects on O$_3$ trends cannot be ignored, particularly in 2022. Shadowed by mid–latitude atmospheric circulation, tropical sea–air coupling, and local land–air feedback processes, a record–breaking super–heatwave event occurred in most cities in eastern China in the summer of 2022, and some cities broke their highest temperature records (Zhang et al., 2023; Zhang et al., 2022). The most important meteorological variables in the MLR model were daily maximum temperature and RH (Tables R1 and R2). The temperature in eastern China showed that the monthly mean nighttime (daytime) temperature in June–August 2022 was 1.0 ℃ (1.1 ℃), 0.8 ℃ (1.4 ℃) and 2.2 ℃ (2.8 ℃) higher than the monthly mean nighttime (daytime) temperature in June–August 2021, respectively (Fig. R3). The monthly mean nighttime (daytime) RH in eastern China in 2022 was 3.2% (3.1%), 1.9% (4.5%), and 9.4% (11%) lower than the monthly mean nighttime (daytime) RH in June–August 2021, respectively. Li et al. (2024) revealed that a sustained heatwave of extremely hot and dry summers in 2022 accelerate photochemical O$_3$ production by increasing anthropogenic and biogenic emissions and exacerbate O$_3$ accumulation by inhibiting dry deposition due to water–starved vegetation, resulting in an increase in O$_3$ pollution by more than 30% in urban areas. Our results also showed an increase in the meteorological components in the 98$^{th}$ and 2$^{nd}$ O$_3$ percentiles in 2022 relative to the meteorological components in the 98$^{th}$ and 2$^{nd}$ O$_3$ percentiles in 2021 (Fig.R2). Therefore, extremely hot and dry weather in 2022 will increase the peak and low O$_3$ concentrations in eastern China, which is probably the main reason for the difference between the May and September 2017–2021 and May–September 2017–2022 meteorological component trends. The above discussion was added to the manuscript, please refer to Page 13 Line 28–32 and Page 14 Line 1–19 in the manuscript.

In addition, the high relative humidity in 2020 and 2021, as well as the impact of COVID–19 pandemic also had a significant impact on the peak and low O$_3$ trend, which is analyzed in detail in **Point 4**.

[Figure]

**Fig.R2** Trends of observed (blue lines), meteorological (red lines), and anthropogenic (red lines) (a) 98th and (b) 2nd O₃ percentiles component in eastern China during May–September 2017–2022. The labels at the top of each panel represent the trend in observed, meteorological, and anthropogenic components.

[Figure]

**Fig.R3** Trends of observed (a) 98th and (b) 2nd O₃ percentiles (blue lines), meteorological (a) 98th and (b) 2nd O₃ percentiles component (red lines) in MLR simulations, and the anthropogenic (a) 98th and (b) 2nd O₃ percentiles component (magenta lines) in eastern China during May–September 2017–2022. The labels at the top of each panel represent the trend in observed, meteorological, and anthropogenic components.

**Table R3**. Observed, Meteorologically, and anthropogenically driven trends of 2% and 98% O$_3$ percentiles in eastern China from 2017 to 2022 and from 2017 to 2021.

| | May–September 2017–2022 trends | | | | | | May–September 2017–2021 trends | | | | | |
| | 2% | | | 98% | | | 2% | | | 98% | | |
| | Obs. | Mete. | Anth. | Obs. | Mete. | Anth. | Obs. | Mete. | Anth. | Obs. | Mete. | Anth. |
|---|---|---|---|---|---|---|---|---|---|---|---|---|
| Total | 0.115 | 0.008 | 0.107 | -0.178 | 0.005 | -0.183 | 0.027 | -0.044 | 0.071 | -0.363 | -0.119 | -0.244 |
| May | 0.322 | 0.017 | 0.305 | -0.020 | -0.661 | 0.641 | -0.438 | -0.257 | -0.181 | -3.702 | -0.968 | -2.734 |
| June | 0.205 | -0.032 | 0.237 | -4.437 | -1.894 | -2.543 | -0.364 | -0.169 | -0.195 | -2.645 | 0.124 | -2.769 |
| July | 0.768 | -0.177 | 0.945 | -1.745 | -1.100 | -0.645 | 0.665 | -0.247 | 0.912 | -2.974 | -2.370 | -0.604 |
| August | 0.371 | 0.084 | 0.287 | -0.687 | -0.156 | -0.531 | -0.003 | -0.260 | 0.257 | -1.473 | -0.908 | -0.565 |
| September | 1.290 | 0.319 | 0.971 | 1.999 | 1.136 | 0.863 | 0.884 | -0.126 | 1.010 | 1.352 | 0.814 | 0.538 |

**Point 4:** The authors have not directly answered why the 2$^{nd}$ percentile O$_3$ increased over 2017–2022. The 2$^{nd}$ percentile should be related to nighttime O$_3$, while the entire manuscript discusses the O$_3$ photochemical formation regime, which is a daytime indicator. More investigation is needed on the nighttime process, such as NO titration of O$_3$, loss of O$_3$ with VOCs, etc.

**Response 4:** We agree with this suggestion. Because vertical profiles of O$_3$ precursors at night are not available due to MAX–DOAS observational limitations, we discuss the possible increase in 2$^{nd}$ O$_3$ percentile based on surface observations and MLR modelling results. It has been analysed in **Point 3** that the extreme hot and dry in 2022 increases the 98$^{th}$ and 2$^{nd}$ O$_3$ percentile concentrations, which is mainly a meteorological effect.

If we look carefully at Fig.R2, which presents the anthropogenic impact for each year on the 98$^{th}$ and 2$^{nd}$ percentile trends, respectively, we will find that the observed increase in 2$^{nd}$ O$_3$ percentile was mainly concentrated after 2020, up to 0.44 ppb/year. The meteorological components did not change significantly in 2020 and 2021 but considerably increased in 2022, with a trend of 0.17 ppb/year 2020–2022. This rapid increase in 2$^{nd}$ O$_3$ percentile is mainly caused by anthropogenic emissions, with a trend of 0.27 ppb/year 2020–2022. Owing to the impact of the COVID–19 pandemic, the decrease in NO$_x$ concentrations was most significant in 2020–2022 (Fig.3b), and the substantial reduction in NO$_x$ concentrations weakened the O$_3$ titration of NO, resulting in an increase in nighttime O$_3$ concentrations, which was also confirmed by the significant negative correlation between the trend of the 98$^{th}$ NO$_2$ percentile and the trend of the 2$^{nd}$ O$_3$ percentiles during May–September 2017–2022 (Fig.4b). A recent study showed that nighttime O$_3$ depletion in China is mainly caused by the wet–scavenging effect and O$_3$ titration from fresh NO emissions (Li et al., 2023). The wet–scavenging effect

was similar to the effect of precipitation, the higher the ambient humidity, the more conducive it was to $O_3$ depletion. The RH at night increased slowly in eastern China during May–September of 2017–2021 (Fig.R3b), and the nighttime RH in 2020 and 2021 was higher than that in other years. Moreover, a general wetting trend has been detected in eastern China during the summer in recent years (Hu et al., 2021). RH had the most significant effect on $2^{nd}$ $O_3$ percentile trends according to MLR results (Table R1). Therefore, the meteorological component had an inhibitory effect on the increase in the $2^{nd}$ $O_3$ percentile trends during May–September 2017–2021. However, owing to the significant emission reduction of $NO_x$ concentrations, the titration of $NO_x$ was weakened, and the decrease in $O_3$ depletion at night led to an increase in the overall $2^{nd}$ $O_3$ percentile trends.

In conclusion, owing to the impact of the COVID–19 pandemic (significant decrease in $NO_x$ concentrations) and unfavorable meteorological conditions (high relative humidity) in 2020 and 2021 in eastern China, the $98^{th}$ $O_3$ percentile concentration in 2020 and 2021 was lower (compared to the $98^{th}$ $O_3$ percentile concentration in 2018 and 2019), while the $2^{nd}$ $O_3$ percentile concentration showed a rapid upward trend. In addition, the extremely hot and dry meteorological conditions in 2022 will increase the $98^{th}$ and $2^{nd}$ $O_3$ percentile concentrations, weakening the decreasing trend in peak $O_3$ concentrations and increasing the upward trend at low $O_3$ concentrations. The above discussion was added to the manuscript, please refer to Page 14 Line 31–34 and Page 15 Line 1–18 in the manuscript.

[Figure]

**Fig.R4**. Trends of surface (a) O₃, (b) NO₂, (c) O₃ exceedance days and O₃ exceedance hours in eastern China during May–September 2017–2022. The red, magenta, and blue solid lines in (a) and (b) indicate the trends for the 98th, 50th, and 2nd percentiles, respectively. The labels on (a) and (b) represent the trends in O₃ and NO₂ for May–September 2017–2022, units: ppb/year. The labels on (c) represent the trends in O₃ exceedance days and O₃ exceedance hours for May–September 2017–2022. The percentage change is indicated in brackets.

[Figure]

**Fig.R5**. Scatterplots showing the relationships between the (a) trend of mean NO₂ concentrations and the trend of 98th O₃ percentiles, (b) trend of 98th NO₂ percentiles and trend of 2nd O₃ percentiles in each city of eastern China during May–September 2017–2022. The correlation coefficients are shown in the top left of each panel, N=number of cities.

**Point 5:** In Section 3.4, although this section should be removed according to my comment #1, conflicts between Figure 9 and Figure S9 are noted. The area proportions presented in Figure 9 are not consistent with the spatial patterns in Figure S9. For example, Figure 9d suggests a NOₓ–limited region up to 75% of the total MLYRP in August of 2022, while Figure S9 shows a NOₓ–limited area smaller than the VOC–limited area. Please check your analysis.

**Response 5:** Thanks for pointing out the conflicts among the figures! We carefully examined the reasons for the conflicting images, which were mainly attributed to differences in data processing. Fig.R6 is based on daily FNR to determine the area proportion of the VOC–limited regime, Transition regime, and NOₓ–limited regime areas, while Fig.R7 is based on monthly mean FNR to determine the area proportion of the VOC–limited regime, Transition regime, and NOₓ–limited regime areas. Fig.R6 shows the trends of TROPOMI observed area proportion for VOC–limited regime, Transition regime, and NOₓ–limited regime over Huaihe river basin during May–September 2018–2022. Here we first determined the O₃ formation

sensitivity based on the daily FNR observed by TROPOMI (FNR <2.1 for VOC–limited regime, FNR>3.2 for $NO_x$–limited regime), and then calculated the area proportion of the daily VOC–limited regime, Transition regime, and $NO_x$–limited regime areas. Finally, the monthly average of the area proportion of daily VOC–limited regime, Transition regime, and $NO_x$–limited regime areas was taken and the trend was calculated, as shown in Fig.R6, the light red dots in (b–d) represent the daily values, and the solid red dots are monthly mean values. Fig.R7 shows the spatial and temporal variations of monthly mean FNR from May–September 2018–2022. the monthly mean FNR was calculated firstly based on TROPOMI observed daily FNR, then threshold value 2.1 and 3.2 was used for monthly mean FNR to determine the $O_3$ formation sensitivity. This section was removed according to **Point 1**.

[Figure]

**Fig.R6**. (a) Variation of monthly mean $O_3$ (~13:30) with monthly mean TROPOMI FNR in HRB during May–September 2022. The solid line represents third–order polynomial fitting. The vertical line represents the maximum value of the fitted curve, and the vertical shadow represents the range of the curve slope from −3 to +3 (transition regime). Trends of TROPOMI observed area proportion for (b) VOC–limited regime, (c) Transition regime, and (c) $NO_x$–limited regime over HRB during May–September 2018–2022. The light red dots in (b–d) represent the daily values, and the solid red dots are monthly mean values.

[Figure]

**Fig.R7**. Spatial and temporal variations of monthly mean FNR from May–September 2018–2022. The date is shown at the top of each panel.

**Specific comments:**

1. Section 2.3, could the authors elaborate more on how trustworthy is the TROPOMI $O_3$ profile retrieval?

Based on **Point 1,** we have deleted section 3.4 (Interannual differences in surface $O_3$ formation sensitivity), and the $O_3$ profile observed by TROPOMI was not employed in the manuscript. Thus, we also removed the description of the TROPOMI $O_3$ profile retrieval in the manuscript.

2. Lines 20–22, Page 8, could the authors say more about why $5^{th}$, $50^{th}$, and $95^{th}$ could represent background, typical, and polluted conditions.

Different percentiles may be related to different influences, such as background concentration levels, emission changes, climate change, and regional transport effects (Lefohn et al., 2010). In principle, the lowest daily $O_3$ concentrations usually occur before sunrise due to nighttime titration of NO, and the low percentile ($2^{nd}$) usually characterizes baseline or background conditions because increases in the low $O_3$ percentile tend to be associated with increases in baseline or background $O_3$ concentrations. Similar conclusions were also obtained from both models and observations (Jacob et al., 1999; Cynthia Lin et al., 2000). $O_3$ pollution

in eastern China generally occurs in the late afternoon on clear days in the warm season (Wang et al., 2022), when the ambient $O_3$ concentration is highest, so the high percentile (98th) characterizes the conditions of the pollution events. The middle percentiles (25th, 50th and 75th) usually follow the same trend as the mean values and therefore represent typical conditions (Cooper et al., 2012; Li et al., 2022). We have added this statement to the manuscript, please refer to Page 8 Line 6–17 in the manuscript.

3. Lines 5–6, Page 12: Why is the primary HCHO contribution much higher than the secondary HCHO at these sites, differing from previous findings cited in the paper? Please provide some explanation.

Atmospheric primary HCHO concentrations are mainly derived from motor vehicle exhaust, petrochemical industry, solvent use, and combustion emissions (Ma et al., 2019). Hefei, Huaibei, and Tai'an are located in the NCP, which is the region with the highest primary emissions of air pollutants in China (Li et al., 2017), The rapid industrialization and urbanization in these developing cities has influenced the primary and secondary HCHO concentrations, and HCHO mainly stems from initial atmospheric pollutants (Lu et al., 2024). Although the articles cited in our manuscript show that secondary HCHO concentrations are much higher than that of primary HCHO, the primary HCHO has also been found to be much higher than secondary HCHO in other cities in China, such as Shenyang (Ma et al., 2019), Lanzhou (Lu et al., 2024), Chengdu (Bao et al., 2022), Nanjing (Hong et al., 2018). Primary HCHO concentrations in Hefei, Huaibei and Tai'an usually reach their maximum in the morning and evening, and vehicle emissions may be the main source of primary HCHO, as industrial emissions do not show a significant diurnal pattern (Hong et al., 2018). We have added this statement to the manuscript, please refer to Page 12 Line 9–14 in the manuscript.

**Technical corrections:**
1. Replace "unbalanced emission reduction in ozone precursors" with $NO_x$ reduction throughout the text?
We have followed this suggestion and carefully checked the manuscript for similar expressions.
2. Line 28, Page 1, remove "experiment".
Thanks for pointing out the unsuitable expression. We have followed this suggestion and corrected the mistake accordingly.
3. Line 25, Page 2, Zhai et al. (2019) is a $PM_{2.5}$ study, not ozone study. Please remove. Also in Line 3, Page 4.

Thanks for pointing out the inappropriate quote, we have corrected the relevant mistakes and carefully checked the manuscript for similar errors.

4. Line 5, Page 3, it should be Li et al. (2020a).

Thanks for pointing out the unsuitable expression, we have corrected the relevant mistakes.

5. Line 22, Page 3, it should be "diagnose".

Thanks for pointing out the unsuitable expression, we have corrected the relevant mistakes.

6. Line 24, Page 8, do not use "one-sided understanding".

We have followed this suggestion and carefully checked the manuscript for similar expressions.

7. In Figure 5, the authors seem to fit the observed monthly $O_3$ and use the fitted lines to connect monthly values. Please replace these with straight lines directly connecting the dots.

Thanks for your suggestion. We have followed this suggestion and replotted the figure. Please refer to Fig.6 in the manuscript.

8. References, journal names should be included.

Thanks for your suggestion. We have followed this suggestion and reorganized the format of the references.

Reference:

[revised manuscript text omitted]

---

## Author Comment (AC2)

**Response to Reviewer 2 Comments**

Thank you for your decision and constructive comments on my manuscript. We have revised the manuscript carefully according to the reviewers' comments. Point–to–point responses are given below. The original comments are black in color, while our responses are in blue. The revised parts in the manuscript are marked in red. All the page number and line number are referred to the revised manuscript.

**Major concerns:**

**Point 1:** How to interpret the variations of low ozone concentrations / background ozone for the targeted region as it's surrounded by polluted areas?

**Response 1:** In principle, the lowest daily $O_3$ concentrations usually occur before sunrise due to nighttime titration of NO, and the low percentile ($2^{nd}$) usually characterizes baseline or background conditions because increases in the low $O_3$ percentile tend to be associated with increases in baseline or background $O_3$ concentrations. Similar conclusions were also obtained from both models and observations (Jacob et al., 1999; Cynthia Lin et al., 2000). Li et al. (2014) reported that concentrations below the $5^{th}$, between $25^{th}$ and $75^{th}$, and above $95^{th}$ represent background, typical, and polluted concentrations, respectively. Thus, in our manuscript, low $O_3$ concentration levels were determined by the $2^{nd}$ percentiles of hourly $O_3$ concentrations in each month for cities in eastern China (Li et al., 2022; Cooper et al., 2012; Gaudel et al., 2020). Based on the $2^{nd}$ $O_3$ percentile concentrations, we explore the driving forces of the low $O_3$ trends in eastern China in recent years. Firstly, the Multiple Linear Regression (MLR) model is used to evaluate the anthropogenic and meteorological contributions to the $98^{th}$ and $2^{nd}$ $O_3$ percentile trends. Then, we discuss the impacts of meteorological factors and anthropogenic emissions on low $O_3$ concentrations, including the impact of the COVID–19 pandemic, and sustained heatwave of extremely hot and dry summer in eastern China in 2022, respectively, and detailed information can be found in the discussion section of the manuscript.

**Point 2:** How to identify the ozone formation sensitivity based on FNR (ratio of formaldehyde and NO₂).

**Response 2:** This method belongs to one of the photochemical indicator diagnostics of $O_3$ formation sensitivity (Chu et al., 2024), and the FNR is the ratio of the volume concentration of HCHO to $NO_2$. HCHO is a transient product reflecting the oxidation of various VOCs, serving as a proxy for VOCs emissions, as used in previous studies (Zheng et al., 2018; Zhang

et al., 2019). The ratio of $NO_2$ to NO in $NO_x$ is relatively constant, and the $NO_2$ concentration can represent the evolution process of atmospheric $NO_x$. Therefore, HCHO and $NO_2$ can be taken as representatives of VOCs and $NO_x$, respectively. This photochemical indicator was originally derived from the photochemical indicator $HCHO/NO_y$ (the ratio of HCHO and $NO_y$ volume concentrations). Jin et al. (2017) suggested that FNR is better suited than $HCHO/NO_y$ for determining $O_3$ formation sensitivity because both HCHO and $NO_2$ have short lifetimes (about a few hours), and their ratios can better represent the competition between OH radicals and the reaction between VOCs and $NO_2$.

This photochemical indicator has been widely used not only in the analysis of ground–based observations, but also in satellite observations, and Martin et al. (2004) were the first to propose the diagnosis of $O_3$ formation sensitivity based on satellite observations of FNR. Currently, this method has been refined and extended to various $O_3$ monitoring instrument products (Ren et al., 2022; Chang et al., 2016), such as OMI (Ozone Monitoring Instrument), TROPOMI (Tropospheric Monitoring Instrument) and GEMS (Geostationary Environment Monitoring Spectrometer).

The reliability of surface FNR in the diagnosis of $O_3$ formation sensitivity was recently assessed by Liu et al. (2021). Based on a joint analysis of multiple in situ observations and model simulations, the validity of FNR in diagnosing $O_3$ formation sensitivity was determined based on the relative changes in $O_3$ generation rates with respect to several indicators. Lin et al. (2022) further extended the FNR to the vertical distribution and diagnosed the $O_3$ formation sensitivity at different altitude using vertical profile of $NO_2$ and HCHO observed by Multi-Axis Differential Absorption Spectrometer (MAX–DOAS), and the methodology employed in our manuscript is consistent with that of Lin et al (2022).

The advantage of this method is that $O_3$ formation sensitivities can be quickly determined from photochemical indicators. However, the threshold value of FNR is always the core problem in the application of this method, and its threshold value will change with different environmental conditions, especially the thresholds in different regions tend to differ greatly. Therefore, there are differences in the thresholds reported in different literatures.

Three steps are involved in determining the $FNR_{sec}$ threshold. First, the surface–hourly averaged secondary HCHO and $NO_2$ VMRs during May–September based on MAX–DOAS observations were normalized by dividing their respective mean values because of the large differences in surface HCHO and $NO_2$ concentrations (Ren et al., 2022). The ratio of the hourly averaged $O_3$ VMRs to the hourly averaged normalized $NO_2$ VMRs ($S_{NO2}$) and the ratio of the hourly averaged $O_3$ VMRs to the hourly averaged normalized secondary HCHO VMRs ($S_{HCHO}$)

were calculated. Finally, third–order polynomials were used to fit $S_{NO2}$ and $S_{HCHO}$ (Fig. R1). When $S_{NO2}$ is significantly larger than $S_{HCHO}$, $O_3$ formation is more sensitive to $NO_x$ (larger $FNR_{sec}$), which is the $NO_x$–limited regime, and vice versa. For example, in Hefei, $S_{NO2}$ and $S_{HCHO}$ intersected at $FNR_{sec}=0.21$. $FNR_{sec}$ less than 0.16 and greater than 0.29 correspond to VOC–limited regime and $NO_x$–limited regime, respectively, where the relative difference between $S_{NO2}$ and $S_{HCHO}$ is more than 25% (Lin et al., 2022), and the range of $FNR_{sec}$ from 0.16 to 0.29 represents a transition regime.

In addition, following the referee's suggestion, the manuscript was edited by Elsevier Language Editing Services (please see the following Elsevier certificate).

[Figure]

**Fig.R1.** Three–order fitting of slopes of $O_3$ VMRs versus normalized $NO_2$ VMRs and slopes of $O_3$ VMRs versus normalized secondary HCHO VMRs in different $FNR_{sec}$ values in Hefei, Huaibei, and Tai'an during May–September based on MAX–DOAS observations. The intersect at $FNR_{sec}$ indicated by the black solid line. The vertical shadow indicates the relative difference between the slopes of $O_3$ VMRs versus normalized $NO_2$ VMRs and slopes of $O_3$ VMRs versus secondary HCHO VMRs within 25% (transition regime). The labels at the top right of each panel represent the intersect $FNR_{sec}$ values and the thresholds for the $NO_x$–limited regime (high) and VOC–limited regime (low) in Hefei, Huaibei, and Tai'an, respectively.

[Figure]

**Fig. R2** Certificate of Elsevier language editing services

**Detailed comments:**

1.Page 1, Line 27: in the main text, you mainly work on "HRB", instead of the whole eastern China. Maybe specifying HRB, instead of eastern China makes more sense.

  Thanks for your suggestion. This study attempts to reveal the surface low and peak $O_3$ trends in eastern Chinese cities in recent years, and to explore the driving forces behind these trends. The surface observations show that the decreased trend in low $O_3$ concentrations and increased trend in peak $O_3$ concentrations are widespread in urban agglomerations in eastern China. In the original manuscript, we first analyzed the trend of low and peak $O_3$ values in the HRB, which were then extended to eastern China. In fact, we conducted MAX–DOAS measurements in three cities in eastern China, namely Hefei, Huaibei and Tai 'an (Fig.R3). First, the three cities are located at similar longitudes with large differences in latitude, transitioning sequentially from south to north. Secondly, the $O_3$ concentrations of the three cities differed greatly, with Tai' an having a higher surface mean MDA8 $O_3$ concentration (82.9 ppb), Hefei having a lower surface mean MDA8 $O_3$ concentration (65.5 ppb), and Huaibei having an intermediate surface mean MDA8 $O_3$ concentration (74.3 ppb). Therefore, Hefei, Huaibei and Tai 'an can be taken as representative of the 105 cities in eastern China to conduct $O_3$ precursor observations. In the new manuscript, we have deleted the analysis of the HRB and reorganized the manuscript.

  The new manuscript mainly includes four parts. First, we report long–term records of surface $O_3$ and related parameters observed at urban air quality monitoring sites and by satellites in eastern China, characterizing the trends of low, typical, and peak surface $O_3$

concentrations during the warm season (May–September) from 2017 to 2022. Then, a Multiple Linear Regression (MLR) model is used to evaluate the anthropogenic and meteorological contributions to the 98th and 2nd $O_3$ percentile trends. Next, secondary formaldehyde (HCHO) and $NO_2$ are employed to diagnose the diurnal variations in $O_3$ formation sensitivity and investigate the reasons for peak $O_3$ concentration trends in the context of current $NO_x$ reduction. Finally, we discuss the reasons for the potential increase in low $O_3$ concentrations and the sensitivity of peak and low $O_3$ trends during the study period. Please refer to our new manuscript for details.

[Figure]

**Fig.R3**. Spatial distributions of surface mean MDA8 $O_3$ concentrations during May–September 2017–2022. The red arrow indicates the name of each city, which are equipped with ground–based MAX–DOAS observations

2. Page 1, Line 29: can you elaborate more on the "typical" ozone concentrations?

Thanks for your suggestion. We have followed this suggestion and elaborated more on the "typical" $O_3$ concentrations. "The decline in typical $O_3$ concentrations is notably slower than that of peak $O_3$ concentrations, approximately -0.02 ppb/year (-0.0% per year) during the same period.". Please refer to Page 1 Line 31–32 in the manuscript.

3. Page 1, Line 31: please rephrase the sentence. "Anthropogenic emissions" is not the "cause" of the ozone trends, maybe "driving force" is better.

Thanks for pointing out the unsuitable expression. We have followed this suggestion and corrected it. Please refer to Page 1 Line 33 in the manuscript.

4. Page 2, Line 3: change "on spatial scales" to "spatially".

Thanks for pointing out the unsuitable expression. We have followed this suggestion and removed relevant expressions base on the new abstract.

5. Page 3, Line 5: duplicate reference of Li et al., 2020a

Thanks for pointing out the inappropriate quote, we have corrected it.

6. Page 5, line 18: have you applied consistent AMF (air mass factor) between MAX–DOAS and TROPOMI for NO$_2$ VCDs?

We did not apply consistent AMF (air mass factor) between MAX–DOAS and TROPOMI for NO$_2$ VCDs. NO$_2$ VCDs observed by TROPOMI was downloaded directly from https://search.earthdata.nasa.gov/search (last access: 1 July, 2024). The AMF calculation for MAX–DOAS is a two–step process that first retrieve the atmospheric aerosol vertical profile and then inputs the aerosol vertical profile into the radiative transfer model SCIATRAN, to compute the atmospheric photon paths that are used to convert the NO$_2$ slanting column concentration into a vertical column concentration (Xing et al., 2017).

7. Page 5, Line 21: why the differences of sensitivity peaks between TROPOMI and MAX-DOAS lead to different HCHO VCD retrievals?

TROPOMI is a space–borne instrument to observe the atmospheric trace gas column concentration from the top down, while MAX–DOAS is a ground–based spectrometer to observe the atmospheric trace gas column concentration from the bottom up. TROPOMI sensitivity peaks in the upper troposphere, which rapidly drops in the atmospheric layers lower than 3 km (Vigouroux et al., 2020), while MAX–DOAS shows an opposite sensitivity that is maximal at the surface and generally becomes negligible above 3 km (De Smedt et al., 2021; Wang et al., 2019). The HCHO concentrations usually concentrated below 2 km over the polluted city (Fig.R4). Therefore, TROPOMI may underestimate the HCHO concentration below 3 km, resulting in a smaller retrieved HCHO VCD. Similar comparative results are also seen in other cities around the world (De Smedt et al., 2021).

[Figure]

**Fig.R4**. Mean profiles of NO₂ and HCHO concentrations in (a) Hefei, (b) Huaibei, and (c) Tai'an during the whole observation period from May to September.

8. Page 5, Line 24-26: so, what's the conclusion? Are MAX-DOAS data not reliable compared to MEE, as the correlation coefficients are not that high (0.66~0.74)? Please specify it.

The MAX–DOAS data are reliable. Although the correlation coefficients are not that high (0.66~0.74), were also comparable to the comparisons reported in previous studies (Lin et al., 2022; Wang et al., 2020). The differences between MAX–DOAS and MEE observations arise from these two components. First, there was a difference in the detection geometries, as the urban NO₂ concentration observed by MAX–DOAS was the result of scanning along a certain direction, whereas the urban NO₂ concentration observed by MEE was sampled in situ. Second, there were some differences in their locations, and the urban NO₂ concentration of the MEE was the average of several in–situ observation stations (10, 3, and 3 in Hefei, Huaibei, and Tai'an, respectively), whereas we only used one MAX–DOAS in each city. We have added this statement to the manuscript, please refer to Page 5 Line 4–10.

9. Page 6, line 27: this is not true for summer, as VOCs can be dominated by biogenic sources.
Thanks for pointing out the unsuitable expression. We have followed this suggestion and removed relevant expressions.

10. Page 6, Line 28: how about transport from surrounding areas? Especially in HRB, the transport cannot be neglected as it's surrounded by the YRD and Jing-Jin-Ji regions. Please elaborate more.
The effect of transport is included in the meteorological factors, and we considered the effect of horizontal transport (wind speed of u and v components) as well as vertical exchange rates

(vertical velocity at 850 hPa) in MLR fitting. The results also show that horizontal winds and vertical exchange rates are also a key meteorological factor affecting $O_3$ concentrations in many cities in eastern China (Table R1 and R2). The effect of horizontal winds can be seen in two aspects, one is that strong winds blow away high concentrations of $O_3$ and its precursors, and the other is that they bring high concentrations of $O_3$ and its precursors. We also discuss this result in detail in the Discussion section of the manuscript as well. Please refer to Page 14 Line 20–30.

**Table R1.** Meteorological drivers of 2% $O_3$ percentile and Pearson correlation coefficient between observed and modeled 2% $O_3$ percentile in each city of eastern China during May–September 2017–2022

| | Meteorological variable | | | R | | Meteorological variable | | | R |
|---|---|---|---|---|---|---|---|---|---|
| | 1st | 2st | 3st | | | 1st | 2st | 3st | |
| Taian | T | RH | U | 0.35 | Beijing | U | RH | T | 0.23 |
| Puyang | T | BLH | RH | 0.50 | Tianjing | U | T | RH | 0.32 |
| Rizhao | U | V | T | 0.46 | Baoding | U | RH | TP | 0.27 |
| Jining | RH | T | V | 0.54 | Lanfang | U | T | V | 0.23 |
| Xinxiang | U | RH | V | 0.41 | Shijiazhuang | RH | BLH | U | 0.27 |
| Jiaozuo | T | RH | U | 0.39 | Handan | RH | V | BLH | 0.20 |
| Heze | RH | T | V | 0.47 | Qinghuangdao | V | U | RH | 0.32 |
| Linyi | RH | U | TP | 0.40 | Cangzhou | BLH | T | MSLP | 0.28 |
| Kaifeng | RH | U | T | 0.49 | Xingtai | RH | BLH | U | 0.17 |
| Zhengzhou | BLH | RH | U | 0.47 | Hengshui | T | V | RH | 0.28 |
| Luoyang | BLH | RH | V | 0.16 | Tangshan | U | V | RH | 0.20 |
| Zaozhuang | RH | T | U | 0.46 | Jinan | V | BLH | RH | 0.59 |
| Lianyungang | RH | V850 | U | 0.37 | Qingdao | V | BLH | RH | 0.26 |
| Shangqiu | RH | T | V | 0.48 | Zibo | V | BLH | RH | 0.59 |
| Xuzhou | RH | BLH | V | 0.48 | Dongying | U | V | V850 | 0.36 |
| Xuchang | BLH | RH | U | 0.53 | Yantai | V | BLH | U | 0.33 |
| Suqian | RH | T | MSLP | 0.40 | Weifang | BLH | T | U | 0.34 |
| Huaibei | RH | U | BLH | 0.59 | Weihai | RH | U | MSLP | 0.36 |
| Pingdingshan | BLH | RH | U | 0.57 | Dezhou | RH | V | BLH | 0.40 |
| Bozhou | RH | V | T | 0.49 | Liaocheng | BLH | V | RH | 0.48 |
| Zhoukou | RH | U | T | 0.49 | Binzhou | BLH | T | V | 0.23 |
| Luohe | RH | U | MSLP | 0.45 | Shaoxing | RH | BLH | T | 0.59 |
| Suzhou | RH | U | BLH | 0.50 | Jinhua | RH | TP | V | 0.60 |
| Huaian | RH | V | TP | 0.42 | Taizhou | V | RH | U | 0.54 |
| Yancheng | V | RH | BLH | 0.37 | Ningbo | RH | V | TP | 0.48 |
| Nanyang | RH | U | BLH | 0.67 | Wuhan | RH | V | BLH | 0.61 |
| Zhumadian | RH | TP | BLH | 0.52 | Changsha | RH | V | BLH | 0.71 |
| Fuyang | RH | V | U | 0.64 | Jinghzou | RH | V | BLH | 0.55 |
| Bengbu | RH | BLH | U | 0.36 | Yueyang | RH | BLH | V | 0.60 |
| Huainan | RH | BLH | U | 0.56 | Zhuzhou | RH | V | BLH | 0.73 |
| Xinyang | RH | TP | T | 0.63 | Xiangtan | RH | V | BLH | 0.72 |
| Suizhou | RH | BLH | U | 0.63 | Yichang | RH | U | V850 | 0.64 |
| Shanghai | RH | V | TP | 0.55 | Yiyang | RH | V | BLH | 0.67 |
| Nanjing | V | RH | T | 0.40 | Changde | V | BLH | RH | 0.59 |
| Wuxi | RH | V | BLH | 0.57 | Jingmen | V | RH | U | 0.63 |
| Changzhou | V | RH | TP | 0.49 | Huangshi | RH | U | V850 | 0.72 |
| Suzhou | RH | V | T | 0.52 | Huanggang | RH | U | V850 | 0.77 |
| Nantong | V | RH | U | 0.62 | Xianning | RH | U | MSLP | 0.74 |

| City | 1st | 2nd | 3rd | R | City | 1st | 2nd | 3rd | R |
|---|---|---|---|---|---|---|---|---|---|
| Yangzhou | V | RH | T | 0.52 | Xiaogan | RH | V850 | BLH | 0.68 |
| Zhenjiang | V | RH | T | 0.55 | Quzhou | V | RH | U | 0.68 |
| Taizhou | V | RH | T | 0.59 | Lishui | RH | V | T | 0.70 |
| Luan | RH | V | T | 0.24 | Wenzhou | RH | V | U | 0.63 |
| Hangzhou | RH | V | BLH | 0.53 | Jiujiang | V | RH | MSLP | 0.58 |
| Jiaxing | RH | V | T | 0.45 | Nanchang | V | RH | V850 | 0.67 |
| Huzhou | V | RH | U | 0.61 | Jingdezhen | BLH | V850 | T | 0.59 |
| Hefei | RH | TP | V | 0.37 | Shangrao | V850 | BLH | RH | 0.65 |
| Wuhu | V | TP | U | 0.46 | Yingtan | BLH | V850 | RH | 0.64 |
| Maanshan | V | U | TP | 0.45 | Yichun | V850 | U | T | 0.63 |
| Tonglin | V | T | TP | 0.50 | Fuzhou | RH | V | BLH | 0.72 |
| Anqing | V | T | TP | 0.56 | Jian | BLH | U | RH | 0.57 |
| Chuzhou | RH | V | BLH | 0.44 | Xinyu | BLH | V | U | 0.62 |
| Chizhou | RH | V | BLH | 0.47 | Pingxiang | V850 | BLH | T | 0.61 |
| Xuancheng | T | RH | U | 0.27 | - | - | - | - | - |

**Table R2.** Meteorological drivers of 98% $O_3$ percentile and Pearson correlation coefficient between observed and modeled 98% $O_3$ percentile in each city of eastern China during May–September 2017–2022

| | Meteorological variable | | | | | Meteorological variable | | | |
|---|---|---|---|---|---|---|---|---|---|
| | 1st | 2st | 3st | R | | 1st | 2st | 3st | R |
| Taian | T | TP | TCC | 0.71 | Beijing | MSLP | BLH | V | 0.30 |
| Puyang | T | V850 | TP | 0.79 | Tianjing | V | MSLP | BLH | 0.27 |
| Rizhao | T | U | TCC | 0.67 | Baoding | MSLP | V | BLH | 0.28 |
| Jining | T | TP | V850 | 0.77 | Lanfang | MSLP | BLH | V | 0.32 |
| Xinxiang | T | TCC | BLH | 0.74 | Shijiazhuang | MSLP | BLH | V | 0.24 |
| Jiaozuo | T | TP | TCC | 0.81 | Handan | T | MSLP | U | 0.31 |
| Heze | T | V850 | TP | 0.78 | Qinghuangdao | TCC | U | MSLP | 0.36 |
| Linyi | T | TP | TCC | 0.73 | Cangzhou | V | BLH | MSLP | 0.35 |
| Kaifeng | T | V850 | TP | 0.78 | Xingtai | MSLP | BLH | V | 0.25 |
| Zhengzhou | T | V850 | TP | 0.78 | Hengshui | V | MSLP | BLH | 0.38 |
| Luoyang | U | TCC | V | 0.28 | Tangshan | TCC | MSLP | T | 0.27 |
| Zaozhuang | T | TP | TCC | 0.81 | Jinan | T | TP | V850 | 0.77 |
| Lianyungang | TCC | U | TP | 0.68 | Qingdao | U | BLH | RH | 0.45 |
| Shangqiu | T | TP | V850 | 0.71 | Zibo | T | TP | V850 | 0.70 |
| Xuzhou | T | TCC | TP | 0.74 | Dongying | T | RH | V | 0.66 |
| Xuchang | T | V850 | TCC | 0.70 | Yantai | U | T | RH | 0.44 |
| Suqian | RH | TCC | TP | 0.74 | Weifang | T | TP | U | 0.68 |
| Huaibei | TCC | T | TP | 0.74 | Weihai | U | T | RH | 0.44 |
| Pingdingshan | T | V850 | U | 0.66 | Dezhou | T | TP | V850 | 0.75 |
| Bozhou | TCC | T | TP | 0.67 | Liaocheng | T | V850 | TP | 0.77 |
| Zhoukou | T | TCC | RH | 0.73 | Binzhou | T | TP | V | 0.66 |
| Luohe | T | V850 | RH | 0.75 | Shaoxing | T | V | TP | 0.66 |
| Suzhou | TCC | T | TP | 0.71 | Jinhua | T | V | TP | 0.59 |
| Huaian | T | TCC | TP | 0.72 | Taizhou | U | V | T | 0.62 |
| Yancheng | TP | V850 | U | 0.56 | Ningbo | T | V | U | 0.69 |
| Nanyang | T | TCC | RH | 0.74 | Wuhan | TCC | RH | V850 | 0.75 |
| Zhumadian | TCC | BLH | TP | 0.67 | Changsha | RH | TCC | V | 0.76 |
| Fuyang | RH | TCC | MSLP | 0.74 | Jinghzou | RH | V850 | T | 0.84 |
| Bengbu | T | TCC | TP | 0.72 | Yueyang | RH | TCC | U | 0.82 |
| Huainan | TCC | RH | MSLP | 0.68 | Zhuzhou | RH | V | T | 0.73 |
| Xinyang | RH | TCC | U | 0.74 | Xiangtan | RH | V | TCC | 0.78 |
| Suizhou | RH | TCC | MSLP | 0.75 | Yichang | RH | TCC | V850 | 0.79 |
| Shanghai | T | U | RH | 0.72 | Yiyang | RH | TCC | V850 | 0.80 |
| Nanjing | TCC | V850 | V | 0.61 | Changde | TCC | V850 | T | 0.74 |
| Wuxi | TP | TCC | U | 0.66 | Jingmen | RH | V850 | T | 0.76 |

| Changzhou | TCC | TP | BLH | 0.60 | Huangshi | RH | TCC | V | 0.76 |
|---|---|---|---|---|---|---|---|---|---|
| Suzhou | T | TP | U | 0.63 | Huanggang | TCC | RH | U | 0.72 |
| Nantong | T | RH | BLH | 0.75 | Xianning | RH | V850 | T | 0.80 |
| Yangzhou | TCC | TP | V850 | 0.61 | Xiaogan | RH | TCC | U | 0.72 |
| Zhenjiang | TCC | TP | MSLP | 0.61 | Quzhou | RH | V850 | T | 0.74 |
| Taizhou | TCC | V850 | U | 0.61 | Lishui | RH | U | V | 0.65 |
| Luan | TCC | MSLP | T | 0.61 | Wenzhou | U | RH | V | 0.65 |
| Hangzhou | TP | TCC | T | 0.69 | Jiujiang | TCC | V850 | RH | 0.80 |
| Jiaxing | T | TP | U | 0.70 | Nanchang | RH | V850 | T | 0.72 |
| Huzhou | TP | TCC | T | 0.67 | Jingdezhen | RH | V850 | TCC | 0.75 |
| Hefei | TCC | TP | V | 0.59 | Shangrao | TCC | V850 | T | 0.76 |
| Wuhu | TCC | TP | V | 0.65 | Yingtan | TCC | V850 | T | 0.78 |
| Maanshan | TCC | TP | MSLP | 0.66 | Yichun | V850 | TCC | T | 0.75 |
| Tonglin | TCC | MSLP | V | 0.63 | Fuzhou | V850 | T | TCC | 0.82 |
| Anqing | TCC | V | TP | 0.58 | Jian | TCC | T | V | 0.66 |
| Chuzhou | TCC | TP | MSLP | 0.58 | Xinyu | V850 | TCC | V | 0.75 |
| Chizhou | TCC | V | MSLP | 0.63 | Pingxiang | V850 | TCC | V | 0.75 |
| Xuancheng | TCC | V | MSLP | 0.58 | - | - | - | - | - |

11. Page 7, Line 7: so, the perturbations of background ozone are neglected when considering the observed O₃ anomalies. It's not true as transport can contribute to this term.

We thank the reviewer for pointing out this issue. We are also aware of this issue, so we have corrected the MLR model in the manuscript. The updated MLR model first de–seasonalizes the $O_3$ concentration time series to remove meteorological variability in $O_3$ concentration. Following Liu et al. (2023) and Li et al. (2020), the MLR model fitted the deseasonalized and detrended 10 d mean 98[th] or 2[nd] $O_3$ percentile time series to the deseasonalized and detrended 10 d mean meteorological variables. The deseasonalized and detrended time series data were constructed by removing the 50 d moving average data from the 10 d moving average data. The stepwise MLR model has the following form:

$$Y(t) = R + \sum_{k=1}^{n} \beta_k X_k(t) \tag{1}$$

Where $Y(t)$ is the deseasonalized and detrended daily surface 98[th] or 2[nd] $O_3$ percentile time series, R is the regression constant, $\beta_k$ is the regression coefficient, and $X_k$ is the deseasonalized and detrended daily meteorological variable considered as a possible $O_3$ covariate. Stepwise regressions were performed, adding and removing terms based on their independent statistical significance to obtain the best model fit.

Daily meteorological variables were obtained from the ERA5 reanalysis data (Download from https://cds.climate.copernicus.eu, last access: January 7, 2024), included temperature (T, °C), surface relative humidity (RH, %), total cloud cover (TCC, %), total precipitation (TP, mm), mean sea level pressure (MSLP, hPa), wind speed of U, V components (U, V, m/s), boundary layer height (BLH, m), and vertical velocity at 850 hPa (V850, m/s).

First, we used the MLR model to remove the effects of meteorological variability from the 2017 to 2022 98th or 2nd $O_3$ percentile trends. We apply Eq. (1) to the meteorological anomalies $X_k$ during May–September 2017–2022, obtained by removing the 6–year means of the 50 d moving averages from the 10 d mean time series. The anomalies calculated in this process were deseasonalized but not detrended. This yields the meteorology–driven 98th or 2nd $O_3$ percentile anomalies $Y_m(t)$

$$Y_m(t) = R + \sum_{k=1}^{n} \beta_k X_k(t) \tag{2}$$

Secondly, to avoid overfitting, only the three most important meteorological parameters were selected based on their individual contributions to the regressed 98th or 2nd $O_3$ percentiles, along with the requirement that they be statistically significant above the 95% confidence level in the MLR model (Li et al., 2018). The fit results and selected meteorological variables varied by city but were regionally consistent (Table R1 and Table R2). The 98th or 2nd $O_3$ percentile anomalies $Y_a(t)$ obtained by deseasonalizing, but not detrending, the 98th or 2nd $O_3$ percentile time series in a similar manner as for the meteorological variables (by removing the 6–year means of the 50 d moving averages). The residual anomaly $Y_r(t)$ after removing the meteorology–driven 98th or 2nd $O_3$ percentile anomalies from the MLR model is given by

$$Y_r(t) = Y_a(t) - Y_m(t) \tag{3}$$

Finally, the residual is an anomalous component that cannot be explained by the MLR meteorological model and is referred to as meteorologically corrected data by (Zhai et al., 2019). It consists of noise due to the limitations of the MLR model and other factors and can be mainly attributed to long–term trends in anthropogenic emission changes over a 6–year period. The trend in the regressed 98th or 2nd $O_3$ percentile reflected the meteorological contribution, and the residual was then used to reflect the presumed anthropogenic contribution. For the updated MLR model, please refer to section 2.4 in the manuscript.

12. Page 8, Line 2: HRB is surrounded by very polluted area, and it can be affected by pollution transport. It's hard to define "background" here, and the background concentrations can change interannually.

We agree with this suggestion, and this is one of the errors in the separation of primary and secondary sources of HCHO by Regression Model. We have extensively reviewed previous studies, and most of the literature has set the background concentration of HCHO at 1 ppb (Sun et al., 2021; Hong et al., 2018; Lin et al., 2022). In addition, according to the ground–level

measurements of HCHO at a rural site in the eastern China by Ma et al. (2016) and Wang et al. (2015), the background level of HCHO near the surface was approximately 1.0 ppb. Therefore, we used this background concentration (1 ppb) in our regression model. Moreover, The fitting results of the regression model are also comparable to those in the previous literature (Hong et al., 2018), so we consider the separation of the primary and secondary sources of HCHO in the manuscript to be reliable.

13. Page 10, line 5: how about the variations in the background?

We have corrected the MLR model in the manuscript (refer to 11). The natural background $O_3$ concentration is deducted before the stepwise multiple regression. Following Liu et al., (2023) and Li et al. (2020), the natural background $O_3$ concentration obtained by the 6–year means of the 50 d moving averages from the 10 d mean time series. The new manuscript focuses on the drivers of trends in peak ($98^{th}$) and low ($2^{nd}$) $O_3$ concentrations.

14. Page 10, line 28: do you include nighttime $O_3$ data in the MLR model? Why?

It is included nighttime $O_3$ data in the MLR model. This study attempts to reveal the surface low, typical, and peak $O_3$ trends in eastern Chinese cities in recent years, and to explore the driving forces behind these trends. The $2^{nd}$ and $98^{th}$ percentiles of hourly $O_3$ concentrations in each month for cities in eastern China were calculated to determine their long–term trends at low and peak concentration levels, respectively (Li et al., 2022; Cooper et al., 2012; Gaudel et al., 2020). In principle, the lowest daily $O_3$ concentrations usually occur before sunrise due to nighttime titration of NO, and the low percentile ($2^{nd}$) usually characterizes baseline or background conditions because increases in the low $O_3$ percentile tend to be associated with increases in baseline or background $O_3$ concentrations.

15. Page 11, Line 2: "(0.108 ppb/year, +114%)", what's this for?

It is the contribution of anthropogenic components on $2^{nd}$ $O_3$ percentiles trends. We updated the MLR model in the manuscript, rearranged section 3.2, and removed this description.

16. Page 11, line 27-28: This is weird. When $S_{no2}$ is much larger than $S_{hcho}$ (low FNR), it should be VOC-limited, instead of NOx-limited.

As shown in Fig.R1, the red and blue line represents the $S_{NO2}$ and $S_{HCHO}$, respectively. When $S_{NO2}$ is much larger than $S_{HCHO}$, the FNR is higher, and the $O_3$ formation is more sensitive to $NO_x$, which is $NO_x$–limited regime.

17. Figure 6: How do you calculate the slope of the ratio of $O_3$ to $S_{no2}$, and the slope of the ratio of $O_3$ to $S_{hcho}$? Are you using the ground measurements for all years? Does each point represent the fitted slope each day? I'm curious the temporal intervals between datapoints shown here.

First, the surface–hourly averaged secondary HCHO and $NO_2$ VMRs during May–September based on MAX–DOAS observations were normalized by dividing their respective mean values because of the large differences in surface HCHO and $NO_2$ concentrations. The ratio of the hourly averaged $O_3$ VMRs to the hourly averaged normalized $NO_2$ VMRs ($S_{NO2}$) and the ratio of the hourly averaged $O_3$ VMRs to the hourly averaged normalized secondary HCHO VMRs ($S_{HCHO}$) were calculated. Finally, third–order polynomials were used to fit $S_{NO2}$ and $S_{HCHO}$ (Fig.R1). $S_{HCHO}$ and $S_{NO2}$ were calculated using data from the warm season (May–September) for all years of MAX–DOAS observations, and each point in Fig.R1 in the manuscript represents the slope for each hour, and the temporal intervals is one hour between datapoints shown in Fig.R1 in the manuscript. We have added this statement to the manuscript, please refer to Page 11 Line 22–27 in the manuscript.

18. Figure 6: I'm curious will the FNR threshold change year by year based on this methodology? How will the interannual variation of the threshold affect your analyses? Please elaborate more.

Differences in the thresholds of the FNR at different times or regions are due to differences in the sources of the corresponding indicator species, whose thresholds vary with environmental conditions (Liu et al., 2021). We calculated the FNR thresholds separately according to different years, and it can be found that the differences in FNR thresholds in different years are very small (FigR5–R7), and the differences in FNR thresholds are mainly between different cities. Therefore, the interannual variation in the threshold have little effect on the results in our manuscript.

[Figure]

**Fig.R5**. Three–order fitting of slopes of $O_3$ VMRs versus normalized $NO_2$ VMRs and slopes of $O_3$ VMRs versus normalized secondary HCHO VMRs in different $FNR_{sec}$ values in Hefei, during (a) May–September 2021–2022, (b) May–September 2021, and (c) May–September 2022.

[Figure]

**Fig.R6**. Three–order fitting of slopes of $O_3$ VMRs versus normalized $NO_2$ VMRs and slopes of $O_3$ VMRs versus normalized secondary HCHO VMRs in different $FNR_{sec}$ values in Huaibei, during (a) May–September 2020–2021, (b) May–September 2020, and (c) May–September 2021.

[Figure]

**Fig.R7**. The same as Fig.R5 but in Tai'an.

---

## Author Response (AR2)

**Response to Reviewer 2 Comments**

Thank you for your decision and constructive comments on my manuscript. We have revised the manuscript carefully according to the reviewers' comments. Point–to–point responses are given below. The original comments are black in color, while our responses are in blue. The revised parts in the manuscript are marked in red. All the page number and line number are referred to the revised manuscript.

**Detailed comments:**

1.Fig. R1. I suggest changing the legend of "slope of the ratio of $O_3$ to normalized $NO_2$" to "ratio of $O_3$ to normalized $NO_2$", because "slope" always represents the slope derived from regression, and here it means the ratio for each hour. It's very confusing. I also suggest revise the y-label to "ratio" too.

**Response 1:** Thanks for your suggestion, following the referee's suggestion, the legend of Fig.R1 was corrected. Please refer to Fig.7 in the manuscript.

[Figure]

**Fig.R1.** Three–order fitting of ratios of $O_3$ VMRs versus normalized $NO_2$ VMRs and ratios of $O_3$ VMRs versus normalized secondary HCHO VMRs in different $FNR_{sec}$ values in Hefei, Huaibei, and Tai'an during May–September based on MAX–DOAS observations. The intersect at $FNR_{sec}$ indicated by the black solid line. The vertical shadow indicates the relative difference between the ratios of $O_3$ VMRs versus normalized $NO_2$ VMRs and ratios of $O_3$ VMRs versus secondary HCHO VMRs within 25% (transition regime). The labels at the top right of each panel represent the intersect $FNR_{sec}$ values and the thresholds for the $NO_x$–limited regime (high) and VOC–limited regime (low) in Hefei, Huaibei, and Tai'an, respectively.

2. Response #7: the sensitivity of TROPOMI always peaks in upper troposphere, and lower in near surface, as represented by averaging kernel. This is true, but it's not fair to say this is the

reason for the underestimation of HCHO concentrations in TROPOMI. Fitting errors, a priori model bias, cloud and aerosols can all contribute to such bias. I suggest authors revising such statement carefully.

**Response 2:** Thanks for your suggestion. We have followed this suggestion and deleted the previous statements, and the corrected expression is "Generally, the $NO_2$ and HCHO VCD observed by TROPOMI were smaller than those observed by MAX–DOAS, and the difference may be caused by fitting errors, a priori model bias, cloud and aerosols, and spatio–temporal resolution." Please refer to Page 4 Line 29–32 in the manuscript.

**Response to Reviewer 3 Comments**

We truly grateful for the reviewers for the valuable and constructive comments, which are very useful for the improvement of the manuscript. We have revised the manuscript carefully according to the reviewers' comments. Point–to–point responses are given below. The original comments are black in color, while our responses are in blue. The revised parts in the manuscript are marked in red. All the page number and line number are referred to the revised manuscript.

**Major comments:**

**Point 1:** MAX-DOAS observations only conducted in three typical cities in eastern China, whether the shift in ozone formation sensitivity from early morning VOC–limited regime to midday NOx–limited regime is widespread in most eastern Chinese cities? Since the satellite observations employed in this study, why not use satellite-derived FNR to diagnose each city's ozone formation sensitivity?

**Response 1:** Thanks for your constructive comments! We have followed the reviewer's comments and use satellite–derived FNR to diagnose each city's ozone formation sensitivity. Owing to the limitations of the observational data, the analysis of diurnal transitions in surface $O_3$ formation sensitivity is limited to three cities in eastern China. Here, other cities in eastern China were further investigated using satellite observations, and we construct conventional FNR using TROPOMI observed $NO_2$ and HCHO VCD from May to September 2018–2022. In order to avoid the misjudgment of $O_3$ formation sensitivity caused by arbitrary selection of FNR thresholds, A third–order polynomial model was applied to investigate the empirical relationship between TROPOMI FNR and surface $O_3$ volume mixing ratios (VMRs, ppb), which has been widely used in other studies (Ren et al., 2022). Since the TROPOMI observed surface $O_3$ VMRs can be obtained after November 2021 in China, we only collected the relationship between TROPOMI FNR and surface $O_3$ VMRs from May to September, 2022. The third–order polynomial fitting relationship between surface $O_3$ VMRs and TROPOMI FNR is shown in Fig. R1a, assuming that the peak of the curve (with a slope of 0) marks the transition from the VOC–limited regime to the $NO_x$–limited regime, the transition regime is defined as a range of slopes between -3 and +3 (Ren et al., 2022). Through the third–order polynomial model, the TROPOMI FNR threshold in eastern China was determined, which are FNR <2.1 for VOC–limited regime, FNR>3.2 for $NO_x$–limited regime (Fig.R1a).

Figure R2 shows the occurrence probabilities of the VOC–limited regime, transition limited regime, and $NO_x$–limited regime spatial distributions derived from TROPOMI observations in eastern China during May–September, 2018–2022. Since the TROPOMI satellite usually transits around 13:30, it can represent the spatial distribution of midday $O_3$ formation sensitivity in eastern China. Apparently, the midday $O_3$ formation sensitivity of most cities in eastern China is under $NO_x$–limited regime, only several cities in the northern part of the NCP and Yangtze River Delta are mainly controlled by VOC–limited regime. In addition, Fig. R1b–d shows the trend of the area proportion of VOC–limited regime, transition regime, and $NO_x$–limited regime in the eastern China, in which the area proportion of VOC–limited regime and transition regime decreases at a rate of 0.62% and 0.18% per year, respectively. While the $NO_x$–limited regime area proportion increased at a rate of 0.80% per year. More importantly, although there is a significant monthly variation in the area proportion of $O_3$ formation sensitivity, it is usually below 50% in May and September, and below 25% in June–August, that is, $NO_x$–limited regime dominates the midday $O_3$ formation sensitivity in eastern China. Due to China's strict control of $NO_x$ emissions in recent years, the surface $O_3$ formation sensitivity in many areas of China has shown a transition from the VOC–limited regime to the transition regime or $NO_x$–limited regime.

In conclusion, significant diurnal transitions in surface $O_3$ formation sensitivity primarily stem from fluctuations in $O_3$ precursors. Early morning conditions (08:00–09:00) are mainly VOC–limited regime, shifting to a $NO_x$–limited regime by midday (12:00–14:00). In addition, the area proportion of VOC–limited regime was also declining, while the $NO_x$–limited regime area proportion was increasing. Consequently, the substantial reduction in $NO_x$ emissions across eastern China has led to pronounced opposite trends in the low (increased) and peak (decreased) surface $O_3$ concentrations, and the surface $O_3$ formation sensitivity to VOCs is generally weakened year by year. Accordingly, the $O_3$ improvement benefits of VOCs emission reduction may become weaker, while the $O_3$ improvement benefits of $NO_x$ emission reduction become larger. Furthermore, the long–distance transport of VOCs has a diminished impact on $O_3$ concentrations due to chemical losses from OH radical oxidation during transport, highlighting $NO_x$ emission reductions as pivotal for intercity and even long–distance efforts to mitigate regional $O_3$ pollution (Wang et al., 2023). We have added this statement to the manuscript, please refer to Page 5 Line 14–18, Page 13 Line 13–34, Page 14 Line 1–13.

[Figure]

**Fig. R1** (a) Variation of monthly mean $O_3$ VMRs (~13:30) with monthly mean TROPOMI FNR in eastern China during May–September 2022. The solid line represents third–order polynomial fitting. The vertical line represents the maximum value of the fitted curve, and the vertical shadow represents the range of the curve slope from −3 to +3 (transition regime). Trends of TROPOMI observed area proportion for (b) VOC–limited regime, (c) Transition regime, and (c) $NO_x$–limited regime over eastern China during May–September 2018–2022. The light red dots in (b–d) represent the daily values, and the solid red dots are monthly mean values.

[Figure]

**Fig. R2** Occurrence probabilities of the (a) VOC–limited regime, (b) transition limited regime, and (c) $NO_x$–limited regime spatial distributions in eastern China derived by TROPOMI observations during May–September 2018–2022.

**Point 2:** The secondary formaldehyde and $NO_2$ (secondary FNR) are employed to diagnose the diurnal variations in ozone formation sensitivity, what is the difference between conventional FNR and secondary FNR?

**Response 2:** We thank the reviewer for pointing out this issue. we used secondary FNR (defined as the ratio of secondary HCHO to $NO_2$; $FNR_{sec} = HCHO_{sec}/NO_2$) in the manuscript as an indicator of $O_3$ formation sensitivity. Compared with conventional FNR (defined as the ratio of HCHO to $NO_2$; $FNR = HCHO/NO_2$), $FNR_{sec}$ eliminate background and primary HCHO interference, improve the accuracy of diagnosing $O_3$ formation sensitivity, and contribute to a better understanding of $O_3$ formation sensitivity (Lin et al., 2022; Xue et al., 2022). The secondary sources in ambient HCHO participated in the photochemical reaction directly, the non–negligible contributions of background and primary HCHO attribute errors to conventional FNR and reduce the accuracy in diagnosing $O_3$ formation sensitivity (Liu, et al., 2021). Hence $FNR_{sec}$ was more favourable for indicating $O_3$ formation sensitivities (Su et al., 2019).

**Detailed comments:**

1. Section 2.4 Stepwise Multiple Linear Regression Model, confusion descriptions, Eq1 : Y(t) is the deseasonalized and detrended daily surface 98th or 2nd ozone percentile time series, the "detrended" is not correct here. While the author use the deseasonalized but not detrended data in the Eq2, it is corrected. The description of Eq1 seems meaningless and may mislead the reader.

**Response 1:** Thanks for your suggestion. We have followed the reviewer's comments and deleted the description of Eq1 in the previous manuscript. Please refer to our new manuscript for details.

2. How do you calculate the Regression Model for source separation in primary and secondary HCHO. Are you using the ground measurements for all years? Are you using the ground measurements in each hour? Please specify it.

**Response 2:** As described in Section 2.5 in the manuscript. CO and $O_x$ ($O_x = O_3 + NO_2$) were selected as tracers to separate the primary and secondary sources of ambient HCHO. The HCHO was measured by ground–based MAX–DOAS, the system was operated only during the daytime (08:00–17:00 local time) with a temporal resolution of 15 min. CO and $O_x$ was collected from the open website of Ministry of Ecology and Environment of China (MEE; https://www.mee.gov.cn; last access: January 7, 2024), and the temporal resolution is one hour. Thus, we first perform hourly averaging of HCHO data from MAX–DOAS observations to

match CO and $O_x$ data from MEE observations. Primary and secondary HCHO will then be separated for all available HCHO data from May to September in the MAX–DOAS measurement period. We have added this statement to the manuscript, please refer to Page 7 Line 13–15 in the manuscript.

3. Page 7 line 20, Are Regression Model reliable? as the correlation coefficients are not that high. Please specify it.

**Response 3:** Thanks for your constructive comments! The Regression Model is reliable. Although the correlation coefficients are not that high (0.55~0.66), were also comparable to the comparisons reported in previous studies (Lin et al., 2022; Sun et al., 2021). As other factors (e.g., meteorological conditions) can also affect the atmospheric HCHO concentration, regression models are difficult to obtain very consistent results. We have added this statement to the manuscript, please refer to Page 7 Line 16–18 in the manuscript.

4. Page 9 line 1-2, How about "typical" ozone concentrations?

**Response 4:** Thanks for your suggestion, the trend in typical $O_3$ concentrations in eastern China from May to September 2017–2022 ranging from -0.4 to 0.3 ppb/year (-0.8–0.8% per year), with about one third of the cities increasing and two thirds decreasing. We have added this statement to the manuscript, please refer to Page 8 Line 34 and Page 9 Line 1–2 in the manuscript.

5. Page 13 line "the relationship between the $O_3$ concentration and $FNR_{sec}$ values from 08:00 to 13:00", why the correlation coefficient of exponential fitting was higher on ozone exceedance days

**Response 5:** This may be due to more dramatic daily variations in $FNR_{sec}$ and ozone concentrations on ozone exceedance days. Fig.R3 (j–l) show the diurnal variation of surface $FNR_{sec}$ during the whole observation in Hefei, Huaibei, and Tai'an, respectively. Fig.R3 (m–o) show the diurnal variation of surface $FNR_{sec}$ during $O_3$ exceedance day in Hefei, Huaibei, and Tai'an, respectively. Compared to the entire observation period, $FNR_{sec}$ on $O_3$ exceedance days exhibits a faster transition from 08:00 to 13:00 and prolonged persistence in the $NO_x$–limited regime. This indicates that the dependence of the $O_3$ production rate on its precursors rapidly shifts with increasing $O_3$ concentration, particularly on $O_3$ exceedance days.

[Figure]

**Fig.R3**. Diurnal variation of surface (a–c) $O_3$ and $NO_2$ VMRs, (d–f) HCHO VMRs contributed by primary and secondary sources, (g–i) the ratio of secondary HCHO to total HCHO VMRs, (j–l) $FNR_{sec}$ during the whole observation, and (m–o) $FNR_{sec}$ during $O_3$ exceedance day in Hefei, Huaibei, and Tai'an during May–September, respectively. The vertical bars in (a–f) represent the one standard deviation. The dot within the box indicates the mean value, the positions of box plots represent the 5th, 25th, 50th, 75th, 95th percentiles, respectively. The horizontal shadow in (j–o) represents the transition regime, the top of the shadow represents the $NO_x$–limited regime, and the bottom of the shadow represents the VOC–limited regime.

6. Page15 line 6-7, can you elaborate more on the "The RH at night increased slowly in eastern China during May–September of 2017–2021 (Fig.10b), and the nighttime RH in 2020 and 2021 was higher than that in other years"

**Response 6:** Thanks for your suggestion, it has been reported in previous studies (Hu et al., 2021), a general wetting trend has been detected in eastern China during the summer in recent years, which is largely related to the increase in summer Precipitation and decrease in summer Potential Evapotranspiration. We have added this statement to the manuscript, please refer to Page 16 Line 1–2 in the manuscript.

7. The discussion should point out the shortcomings in this study and future research perspectives.

**Response 7:** Thanks for your constructive comments! We have pointed out the shortcomings in this study in discussion section. "Owing to the limitations of the observational data, the analysis of surface $O_3$ precursors and $O_3$ formation sensitivity is limited to three cities in eastern China. Although other cities in eastern China were further investigated using satellite observations, TROPOMI only provides observation results for column concentrations at approximately 13:30 each day, which did not allow us to obtain diurnal variations in the $O_3$ formation sensitivity. Further observations must be extended to southern and coastal cities to investigate the relationship between $O_3$ and its precursors more comprehensively." We have added this statement to the manuscript, please refer to Page 16 Line 7–11 in the manuscript.

**Reference:**

Hu, W., She, D., Xia, J., He, B., and Hu, C.: Dominant patterns of dryness/wetness variability in the Huang-Huai-Hai River Basin and its relationship with multiscale climate oscillations, Atmos. Res., 247, 105148, 2021.

Liu, J., Li, X., Tan, Z., Wang, W., Yang, Y., Zhu, Y., Yang, S., Song, M., Chen, S., Wang, H. and Lu, K.: Assessing the Ratios of Formaldehyde and Glyoxal to NO2 as Indicators of O3–NO x–VOC Sensitivity. Environ. Sci. Technol., 55(16), pp.10935-10945, 2021.

Lin, H., Xing, C., Hong, Q., Liu, C., Ji, X., Liu, T., Lin, J., Lu, C., Tan, W., Li, Q., and Liu, H.: Diagnosis of Ozone Formation Sensitivities in Different Height Layers via MAX-DOAS Observations in Guangzhou, J. Geophys. Res-atmos., 127, 10.1029/2022jd036803, 2022.

Ren, J., Guo, F., and Xie, S.: Diagnosing ozone–NOx–VOC sensitivity and revealing causes of ozone increases in China based on 2013–2021 satellite retrievals, Atmos. Chem. Phys., 22, 15035-15047, 10.5194/acp-22-15035-2022, 2022.

Su, W., Liu, C., Hu, Q., Zhao, S., Sun, Y., Wang, W., Zhu, Y., Liu, J. and Kim, J: Primary and secondary sources of ambient formaldehyde in the Yangtze River Delta based on Ozone Mapping and Profiler Suite (OMPS) observations. Atmos. Chem. and Phys., 19(10), pp.6717-6736, 2019

Sun, Y., Yin, H., Liu, C., Zhang, L., Cheng, Y., Palm, M., Notholt, J., Lu, X., Vigouroux, C., Zheng, B., Wang, W., Jones, N., Shan, C., Qin, M., Tian, Y., Hu, Q., Meng, F., and Liu, J.: Mapping the drivers of formaldehyde (HCHO) variability from 2015 to 2019 over eastern China: insights from Fourier transform infrared observation and GEOS-Chem model simulation, Atmos. Chem. Phys., 21, 6365-6387, 10.5194/acp-21-6365-2021, 2021.

Wang, Y., Jiang, S., Huang, L., Lu, G., Kasemsan, M., Yaluk, E. A., Liu, H., Liao, J., Bian, J., Zhang, K., Chen, H., and Li, L.: Differences between VOCs and NO$_x$ transport contributions, their impacts on O$_3$, and implications for O$_3$ pollution mitigation based on CMAQ simulation over the Yangtze River Delta, China, Sci. Total Environ., 872, 10.1016/j.scitotenv.2023.162118, 2023.

Xue, J., Zhao, T., Luo, Y., Miao, C., Su, P., Liu, F., Zhang, G., Qin, S., Song, Y., Bu, N., and Xing, C.: Identification of ozone sensitivity for NO$_2$ and secondary HCHO based on MAX-DOAS measurements in northeast China, Environ. Int., 160, 10.1016/j.envint.2021.107048, 2022.